# Arabidopsis γ-H2A.X-INTERACTING PROTEIN participates in DNA damage response and safeguards chromatin stability

Tianyi Fan[1,4], Huijia Kang[1,2,4], Di Wu[1,4], Xinyu Zhu[3], Lin Huang[1], Jiabing Wu[1] & Yan Zhu [1] ✉

Upon the occurrence of DNA double strand breaks (DSB), the proximal histone variant H2A.X is phosphorylated as γ-H2A.X, a critical signal for consequent DSB signaling and repair pathways. Although γ-H2A.X-triggered DNA damage response (DDR) has been well-characterized in yeast and animals, the corresponding pathways in plant DDR are less well understood. Here, we show that an Arabidopsis protein γ-H2A.X-INTERACTING PROTEIN (XIP) can interact with γ-H2A.X. Its C-terminal dual-BRCT-like domain contributes to its specific interaction with γ-H2A.X. *XIP*-deficient seedlings display smaller meristems, inhibited growth, and higher sensitivity to DSB-inducing treatment. Loss-of-function in *XIP* causes transcriptome changes mimicking wild-type plants subject to replicative or genotoxic stresses. After genotoxic bleomycin treatment, more proteins with upregulated phosphorylation modifications, more DNA fragments and cell death were found in *xip* mutants. Moreover, XIP physically interacts with RAD51, the key recombinase in homologous recombination (HR), and somatic HR frequency is significantly reduced in *xip* mutants. Collectively, XIP participates in plant response to DSB and contributes to chromatin stability.

Maintaining genome integrity is essential to all the organisms for growth and development, and ensures the transmission of genetic information across generations. Unrepaired DNA damage remaining in cell cycle progression will ultimately lead to gene mutation, chromatin instability, and even cell death[1]. Rapid and accurate response to DNA damage-inducing stress is thus crucial for preserving the fidelity of genetic information, especially in the rapidly proliferating cells in plant shoot meristems, from which post-embryonic germ cells are derived. DNA double strand break (DSB) has been widely considered as the most cytotoxic and mutagenic form of DNA damage, and can be elicited by exogenous sources such as ionizing radiation or endogenous events such as DNA replication[2]. To eliminate DSBs, eukaryotic cells deployed a variety of proteins to the damaged chromatin region. Some

directly participate in DNA repair, while others trigger necessary signaling pathway, including cell cycle progression delaying, to coordinate necessary repair process. Collectively, these events comprise the DNA damage response (DDR)[3].

Plant DDR is usually orchestrated by two phosphoinositide-3-kinase-related protein kinases, ATAXIA TELANGIECTASIA MUTATED (ATM) kinase and ATM AND RAD3-RELATED (ATR) kinase, which act as master regulators via triggering phosphorylation signaling cascades[4]. ATM is primarily responsive to DSBs, and is required for the transcriptional induction in response to ionizing radiation in Arabidopsis. ATR is mainly involved in replication stress response[5]. H2A.X, a conserved histone variant of H2A, functions as a key player in damage signaling[6]. Upon DSB occurrence, H2A.X in proximity to DSB site is

[1]State Key Laboratory of Genetic Engineering, Collaborative Innovation Center for Genetics and Development, Department of Biochemistry, Institute of Plant Biology, School of Life Sciences, Fudan University, Shanghai 200438, China. [2]Ministry of Education Key Laboratory for Biodiversity Science and Ecological Engineering, Institute of Biodiversity Science, School of Life Sciences, Fudan University, Shanghai 200438, China. [3]Department of Chemical Engineering (Tanwei College), Tsinghua University, Beijing, China. [4]These authors contributed equally: Tianyi Fan, Huijia Kang, Di Wu. ✉e-mail: zhu_yan@fudan.edu.cn

quickly phosphorylated by ATM and ATR kinases on the C-terminus (corresponding to S139 in Arabidopsis) as γ-H2A.X[7]. Such phosphorylation is widely accepted as a very early event of DNA damage response highly conserved in eukaryotes, and in mammals it can occur within minutes after DSB induction and extend to a few megabases around the site of DNA damage. γ-H2A.X is the major signal to recruit signaling and repair factors to the DNA broken site, thereby promoting efficient repair of damaged chromatin[8]. Loss of H2A.X, or deficiency in H2A.X phosphorylation (i.e., γ-H2A.X), will compromise genomic integrity, reduce DSB repair, and eventually lead to genomic instability, radio-sensitivity, and tumor susceptibility in mammalian cells[6,9]. Arabidopsis genome has two redundant genes encoding H2A.X. Although dispensable for plant viability and fertility, the mutants and silenced lines of *H2A.X* have been reported as mildly hypersensitive to genotoxin and defective in DSB repair[10–13].

A folding unit of approximately 95 residues was originally identified in tumor suppressor protein BREAST CANCER ASSOCIATED 1 (BRCA1) and designated as BRCA1 Carboxyl-Terminal (BRCT) domain. It consists of a four-stranded parallel β-sheet surrounded by three α-helices, and acts as a phospho-protein-binding domain[14]. More BRCT-like domains have been progressively identified in a variety of proteins implicated in DNA metabolism and repair, and are thus proposed to play important biological roles via mediating phospho-protein recognition[15]. Based on the sequence similarity and resolved structure, they can be further subdivided into 9 families as a clan in the public protein database Pfam.

Mammalian MEDIATOR OF DNA DAMAGE CHECKPOINT PROTEIN-1 (MDC1) are huge proteins comprising several domains mediating protein-protein interaction, including tandem BRCT-like domains in its C-terminus. Such tandem BRCT-like domains showed high specificity and strong affinity to the phosphorylated C-terminus of γ-H2A.X but not the unmodified form, thus allowing MDC1 to act as a specific sensor for γ-H2A.X in mammalian cells. After docking on γ-H2A.X, MDC1 recruits further key DDR proteins, including more ATM kinase, to the DNA damage site, thus implicating MDC1 as a DSB mediator between the early DNA damage response and the subsequent signaling and repair[15–18]. In consistence, downstream ATM signaling events were found defective in the absence of MDC1, underscoring the critical role of MDC1 in controlling proper DNA damage response and maintaining genomic stability. *MDC1*-knockout mice showed chromosomal instability, defects in DSB repair, radio-sensitivity, and cancer predisposition[18]. Notable, these phenotypes were obviously similar to those observed in mice lacking histone H2A.X[19], reflecting the highly relevant roles of MDC1 and γ-H2A.X in mediating DSB signaling.

In plants, DDR plays a critical role in genomic stability, cell cycle control and meristem maintenance[20]. Although ATM/ATR and their kinase activities are conserved in plants, there is significant divergence between downstream DDR pathways in plants and animals, largely resulted from the missing of clear plant homologs of key animal DDR factors in signaling cascade, such as the γ-H2A.X reader MDC1 and many crucial checkpoint kinases[21]. Notably, the specific γ-H2A.X reader has not been characterized in plants yet, and thus the biological significance of DNA damage-induced H2A.X phosphorylation remains unclear, and its role in maintaining plant genome integrity in response to genotoxic stresses awaits to be elucidated.

## Results

### Identification of plant γ-H2A.X-interacting protein

We synthesized a 15-amino acid-long peptide (named as p1) highly specific to Arabidopsis H2A.X C-terminus (residues 128-142), and its derivative peptide p2 with residue S139 phosphorylated. Both peptides were N-terminally biotinylated so as to be immobilized by Streptavidin-coated magnetic beads (Fig. 1a), and were used to retrieve interacting proteins from Arabidopsis protein extracts in peptide pulldown followed by Mass Spectrum (ppd-MS) analysis. In this study,

only one protein encoded by At4g03130 interacted specifically with phosphorylated peptide p2 but not p1 (Fig. 1b and Supplementary Data 1), indicative of its specific recognition of γ-H2A.X but not H2A.X. We hence designated this Arabidopsis protein as γ-H2A.X-interacting protein (XIP).

### XIP isoforms with difference in the C-terminal BRCT-like domains

There have been four transcript isoforms of At4g03130 gene in the public database. In addition to the reviewed full-length transcripts NM_001340429.1 and NM_0013404301.1, the other two are both partial transcripts lacking of 3′-terminus, and have a negligible shift in the selection of start codon (with a slight difference in N-terminal 4 residues). The intact 3′-end was determined in this study by using rapid amplification of cDNA end (RACE) method. It extended into the end of opposite neighboring gene At4g03120 and formed a large "tail to tail" overlap (Fig. 1c). In comparison with the above-mentioned full-length transcripts, the sequenced transcript by RACE differs mainly in the extension of the 3′-end, thus encoding protein product with a longer C-terminus. Hereinafter, we named these encoded protein isoforms, based on their molecular weight, as XIP-L (RACE product), -M (NP_001329384.1), and -S (NP_001329385.1), respectively (Fig. 1c, d).

All the encoded XIP isoforms contain a significant 'RTT107_BRCT_5' domain (the fifth domain of REGULATOR OF TY1 TRANSPOSITION PROTEIN 107, PF16770), one family of BRCT-like domains (Pfam 34.0, http://pfam.xfam.org). Notably, the "significance" is labelled by Pfam and means the high similarity to the founding domain (e.g., RTT107_BRCT_5) identified by Pfam. Besides, XIP-L and XIP-M also harbor an additional C-terminal insignificant BRCT domain (BRCA1 C-Terminus, PF00533) of different length. These two domains both adopt a similar compact α/β fold, thus constituting a dual-BRCT-like domain (Fig. 1d and Supplementary Fig. 1). The dual-BRCT-like domain in XIP can also be found in the C-terminus of human MDC1, which is crucial for its interaction with γ-H2A.X. However, at the primary protein sequence level, all the XIP isoforms, even the largest XIP-L, had an identity of no more than 25% with the animal MDC1 proteins only within the BRCT-like domain(s) (Supplementary Fig. 1), without any significant similarity found in the rest of these proteins (Supplementary Fig. 2). Notably, the dual-BRCT-like domain in XIP can be found in both middle and C-terminus of human PAX-INTERACTING PROTEIN 1 (PAXIP)/PAX TRANSCRIPTION ACTIVATION DOMAIN-INTERACTING PROTEIN (PTIP), which also comprises the fifth BRCT-like domain in its N-terminus. The N-terminal and the middle BRCT-like domains of PAXIP/PTIP also interact with other phospho-epitope in many proteins, including the well-studied animal-specific P53 BINDING PROTEIN 1 (53BP1) and PAX2, linking PAXIP/PTIP activity to multiple aspects of human cellular response to DNA damage[22,23].

### Dual- but not single-BRCT-like domains mediate the interaction of XIP with γ-H2A.X

The C-termini of three XIP protein isoforms were constructed as GST-fused recombinant proteins for peptide-pulldown (ppd) verification. In addition to the above retriever peptides p1 and p2, we synthesized another peptide p3, in which S139 was replaced by glutamic acid, mimicking the negatively-charged phosphorylated serine (Supplementary Fig. 3a). Our ppd experiment showed that both dual-BRCT-like domains (XIP-L and -M), but not single-BRCT-like domain (XIP-S), specifically interacted with peptide p2 but not p1 (Supplementary Fig. 3b), indicating that the second insignificant BRCT domain is indispensable to the first RTT107_BRCT_5 domain in γ-H2A.X recognition. Besides, both of these dual-BRCT-like domains also efficiently bound p3 peptide, suggesting that they preferred to the C-terminus of γ-H2A.X carrying negative charge rather than the specific phospho-epitope.

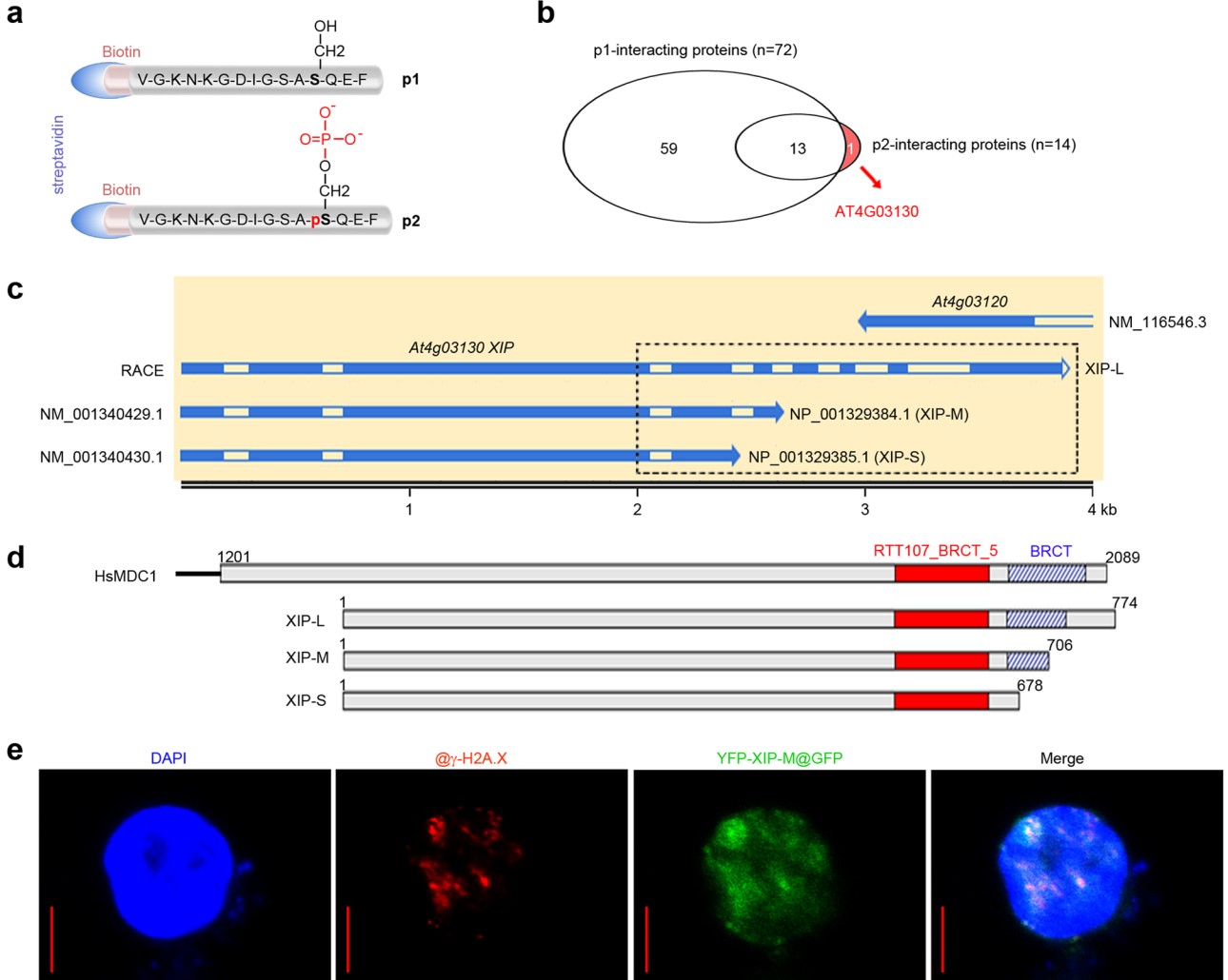

**Fig. 1 | XIP was identified as the specific interacting protein of γ-H2A.X.**
**a** Schematic diagram of N-terminally biotinylated peptides p1 and p2. **b** Venn diagram of the interacting proteins retrieved from p1 and p2. The p2-specific interacting protein, AT4G03130, was shown in red. **c** The transcript isoforms of *XIP*. The dotted box indicates the different C-termini comprising BRCT-like domain(s). **d** The alignment of XIP-L, -M and -S proteins encoded by different transcript isoforms of *XIP* with HsMDC1. RTT107_BRCT_5 and BRCT domains are marked in red (solid) and blue (striped), respectively. **e** The localization of YFP-XIP-M and γ-H2A.X in immuno-staining observation. XIP-M-expressing protoplasts were challenged by 1 μM BLM for 3 hours and immuno-stained by using GFP and γ-H2A.X antibodies. DAPI was used for nuclei staining. Representative image from 9 well-stained protoplasts was shown. Bar=5 μm.

Reverse transcription-quantitative PCR (RT-qPCR) result showed that NM_001340429.1 was the major transcript isoform of *XIP* in all the examined plant organs (Supplementary Fig. 4a, b). In comparison, the transcript level of NM001340430.1 (encoding single-BRCT-like domain) constitutively remained at a low level. Considering the strong correlation of γ-H2A.X signaling pathway with DSB damage, we also challenged seedlings with hydroxyurea (HU) which triggered replication fork collapse, a common source of DSB in vivo, and radiomimetic reagent bleomycin (BLM) which directly caused DSB. The transcript levels of three isoforms were not significantly affected by exogenous replicative or genotoxic stresses (Supplementary Fig. 4c). We then expressed and purified the soluble His-SUMO-tagged recombinant protein XIP-M, the protein product of the major transcript isoform of *XIP*, for the ppd verification. Again, the full-length XIP-M interacted specifically with both p2 and p3 retrievers, but not p1 (Supplementary Fig. 3c), suggesting that dual-BRCT-like domain contributes to the specificity of XIP recognition to γ-H2A.X.

We expressed YFP-fused XIP-M in mesophyll protoplasts (Supplementary Fig. 5). Under the mock condition, YFP signal was obviously localized in both cytoplasm and nuclei. In contrast, shortly after BLM treatment (30 mins), XIP-M was found enriched in the nuclei in most observed protoplasts, suggesting XIP-M was translocated into nuclei in response to genotoxic stress. After longer BLM treatment (3 h), the immunostaining assay revealed that, in contrast to the largely dispersed distribution under mock condition (Supplementary Fig. 6), the XIP signals became focused and co-localized well with all the observed BLM-induced γ-H2A.X foci in DAPI-stained nuclei, although with a slight diffusion around (Fig. 1e).

**Characterization of *xip* mutant**

We identified a T-DNA insertion mutant of *XIP* with insertion in gene body (FLAG_589H07, Wassileskija (Ws) ecotype) and hereinafter named it as *xip* in this study. This mutant was verified as a functional knock-out mutant by RT-qPCR and RACE analyses (Supplementary Fig. 7a, b). Compared to wild type (WT, Ws ecotype), mutant seedlings deficient in *XIP* showed slightly smaller organs under normal growth conditions (Fig. 2a). In consistence, we observed smaller meristem zone in *xip* root tips (Fig. 2b and Supplementary Fig. 8). Both HU and BLM treatments significantly inhibited seedling growth, the root meristem zones within root tips, and primary root elongation

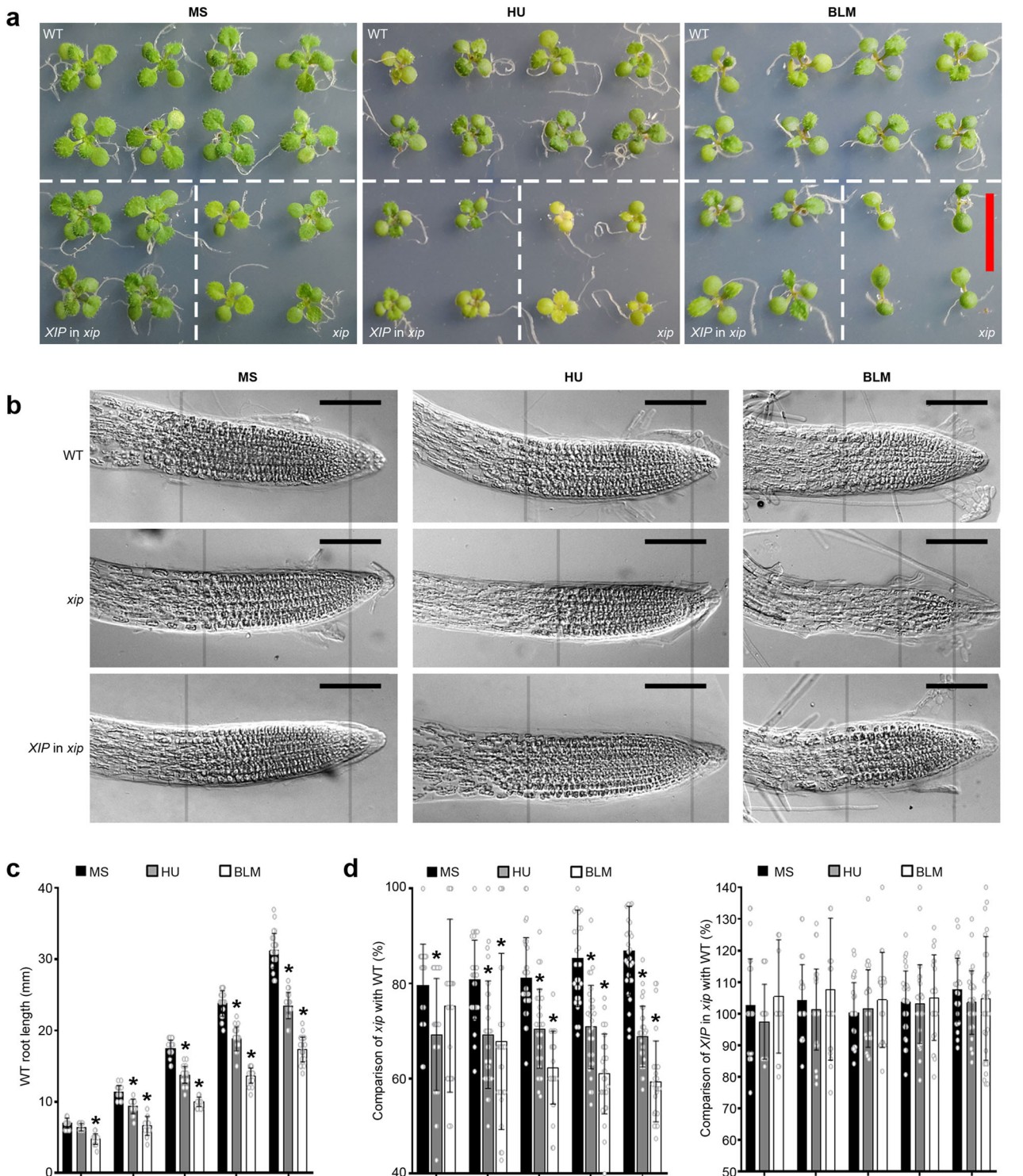

**Fig. 2 | Growth inhibition of *xip* mutant and its sensitivity to genotoxic stress.** **a** 12-day-old WT and *xip* mutant germinated and grown on MS medium with or without genotoxin (1 mM HU or 2.5 μM BLM). Bar = 10 mm. **b** Differential interference contrast (DIC) images taken on roots at 6 days after germination (DAG). The grey lines were added to facilitate the comparison of meristems, in which cells did not enlarge or elongate as revealed by DIC. Representative image from 20 root tips was shown. Bar = 50 μm. **c** Comparison of the primary root elongation in WT germinated and vertically grown on MS medium with or without genotoxin (1 mM HU or 2.5 μM BLM). Mean values of WT root length were shown together with error bars indicating ±SD from 30 biologically independent plants. Asterisks indicated

the statistically significant differences between the MS medium and genotoxin treatment ($p < 0.05$, *t*-test, two-tail). All the *p* values can be found in the source data. **d** Comparison of the primary root elongation in *xip* mutant (left panel) or *XIP* in *xip* (right panel) with WT (shown as percentages) germinated and vertically grown on either MS medium or medium containing 1 mM HU or 2.5 μM BLM. Mean values of the relative root length (percentages) were shown together with error bars indicating ±SD from 30 biologically independent plants. Asterisks indicated the statistically significant differences between the MS medium and genotoxin treatment ($p < 0.05$, *t*-test, two-tail). All the *p* values can be found in the source data.

(Fig. 2a–c and Supplementary Fig. 8). The inhibition became more severe in *xip* (Fig. 2a–d), indicative of the hypersensitivity of *xip* seedlings to DSB stress. Till now, the biological significance of each transcript isoform cannot be fully elucidated (Supplementary Fig. 4). Therefore, we cloned the entire genomic DNA of *XIP* (including 2 kb of promoter and 1 kb of terminator of the longest *XIP-L*) and transformed it into *xip* (named as *XIP* in *xip*), which complemented the growth inhibition of *xip* and recovered the WT-like root meristem zone (Fig. 2a–d and Supplementary Fig. 8). These results indicated that *XIP* functions in both the maintenance of meristematic activity and the plant resistance to replicative and genotoxic stresses.

## Multiple genes were mis-regulated in *xip* mutant
Considering the inhibited growth of *xip* mutant and its hypersensitivity to genotoxic stresses, we examined the transcript levels of several well-studied marker genes in cell cycle progression and several key genes implicated in DSB signal transduction and DNA repair (Fig. 3a, b).

HU treatment in this study resulted in the significantly elevated phase-specific *H3.1* and *CYCB1;1*, indicative of the arrest of S- and $G_2$-to-M phases in cell cycle progression, whereas the relevant DNA repair genes were not significantly activated. The HU-activated *H3.1* and *CYCB1;1* was stronger in *xip* mutant when compared to WT ($p < 0.05$, *t*-test) (Fig. 3a), suggesting that *xip* mutant displayed higher sensitivity to replication fork stress. BLM treatment significantly activated DNA damage genes *WEE1*, *PARP2*, *RAD51* and *RAD54* but not sensor kinase genes *ATM* and *ATR*, as well as cell cycle gene *CYCB1;1* but not *H3.1*, underscoring a different transcriptional response. In this case, *CYCB1;1* transcript level was comparable in WT and *xip* mutant, however, all four examined BLM-induced DNA repair-related genes were not full activated in *xip* mutant as compared to WT (Fig. 3b), suggesting that the full activation of these key genes in the DDR pathway is dependent on XIP activity.

DDR causes massive transcription reprogramming. We thus performed a comprehensive RNA-seq analysis on both WT and *xip* seedlings under mock, HU and BLM treatments, respectively. Among these transcriptome data, we were concerned with the significant transcriptional changes caused by replication or genotoxin stress (marked as HU/Mock or BLM/Mock); as well as those *XIP*-dependent transcriptional changes under the same condition (mock or genotoxin, marked as *xip*/WT). After normalization, those genes with both fold change of transcript level ≥2 and *p*-value ≤0.05 were considered as differentially expressed genes (DEGs) (Supplementary Fig. 9 and Supplementary Data 2). Among them, we noticed that the DEG number in *xip* after BLM treatment (*xip*-BLM/*xip*-Mock) was obviously smaller (338 up- and 144 downregulated genes) when compared to those in WT after BLM treatment (WT-BLM/WT-Mock, 771 up- and 742 downregulated genes), suggesting that the transcriptional response to BLM challenge may be less severe in *xip* mutant when compared with that in WT (Supplementary Fig. 9), which was in line with our quantitative RT-qPCR examination (Fig. 3b and Supplementary Fig. 10), and was further supported by the principal component analysis (PCA) (Supplementary Fig. 11).

The DEGs were consequently analyzed in the following gene ontology (GO) analysis (HU in Fig. 3c and BLM in Fig. 3d, respectively). In this study, the overrepresentation in 'DNA repair' (GO: 0006281) was specifically found in WT-BLM/WT-Mock. In comparison, the significant upregulated DEGs in WT after HU treatment (marked as WT-HU/WT-Mock) had a clear overrepresentation in 'response to stress' (GO: 0006950), and also many other related categories, such as 'response to biotic stimulus' (GO: 0009607) and 'response to oxidative stress' (GO: 0006979). Among these categories, only the overrepresentation in 'response to stress' and 'response to oxidative stress' was still found in WT-BLM/Mock, although with much less significance and much smaller gene number. Intriguingly, these enriched GO categories were also clearly found in the upregulated DEGs in *xip* compared to WT

under mock condition (*xip*-Mock/WT-Mock). In the most enriched GO category 'response to stress (GO: 0006950)', we noticed that 132 out of the 264 upregulated DEGs (50%) in *xip* mutant were overlapped with those in WT-HU/WT-Mock and WT-BLM/WT-Mock (Supplementary Fig. 12), supporting that *XIP*-depletion partially mimics the WT plants under replication and/or genotoxin stresses in activating diverse responsive genes. In consistence, the GO enrichments involved in diverse 'responses' in WT-HU/WT-Mock and WT-BLM/WT-Mock were obviously attenuated in the *xip* mutant background (*xip*-HU/*xip*-Mock and *xip*-BLM/*xip*-Mock, respectively) (Fig. 3c, d).

We also noticed that these enriched GO categories involving diverse responses are also found in the upregulated genes in *xip*-BLM/WT-BLM (with BLM treatment, *xip* compared to WT), although with less significance. Intriguingly, these GO categories were obviously absent in the upregulated genes in *xip*-HU/WT-HU (with HU treatment, *xip* compared to WT) (Fig. 3d), indicating that these enriched GO categories of upregulated genes caused by *XIP*-depletion have been effectively offset predominantly by the exogenous HU treatment, and to a less extent, by BLM challenge. Taken together, *XIP* was highly implicated with the transcription reprogramming triggered by replication and/or genotoxin stresses.

## Phospho-proteomic analysis reveal changes in the DDR pathway
DDR involves a complicate network for the coordination of DNA repair, and the signaling cascades rely heavily on protein phosphorylation by serine/threonine kinases including ATM and ATR[4,24]. We noticed that the category of phosphorylation/protein phosphorylation was overrepresented in the upregulated DEGs in *xip*-Mock/WT-Mock (Fig. 3d), suggesting that multiple genes involved in the homeostasis of protein phosphorylation were abnormally upregulated in *xip* mutant.

Considering the direct effect of BLM in DSB induction, we extracted histones from WT and *xip* mutant grown under either mock conditions or BLM treatment (four protein samples) and examined the content of γ-H2A.X by using specific antibody[25]. The γ-H2A.X signal was negligible in both WT and *xip* under mock condition, but was detectable in WT and more obvious in *xip* after BLM treatment (Fig. 4a), suggestive of the presence of more DSB in *xip* mutant in response to genotoxin stress. Such finding also indicated that XIP facilitates the reduction of γ-H2A.X.

Next, we extracted proteins for phospho-peptide enrichment and employed a label-free quantification (LFQ) phospho-proteomics approach to systematically survey the relevant modification changes. To prevent the unexpected bias resulted from the changed protein abundance in different samples, in addition to the enriched phospho-peptides (Supplementary Data 3), we also analyzed the proteome by using quantitative mass spectrometry (Supplementary Data 4), and normalized the phospho-peptides by the abundance of corresponding proteins (Supplementary Data 5). Those proteins harboring serine/threonine residue with up- or downregulated phospho-modification in fold change ≥2 and *p* < 0.05 were considered as differentially phosphorylated proteins (DPPs) (Fig. 4b and Supplementary Data 5). We found 37 upregulated DPPs in WT-BLM/WT-Mock in this study, including the best-studied γ-H2A.X (Supplementary Fig. 13). In comparison, the number of upregulated DPPs dramatically increased to 218 in *xip* mutant. GO analysis showed that, except for the phosphorylation-related categories, most of them were responsive genes implicated in stimulus response, signal transduction, plant development (Fig. 4b). There was no significant GO enrichment of 24 upregulated DPPs in *xip*-Mock /WT-Mock, while a much higher number of upregulated DPPs (*n* = 193) was found in *xip*-BLM/WT-BLM. The GO enrichment of DPPs in *xip*-BLM/WT-BLM was largely similar to that in *xip*-BLM/*xip*-Mock (Fig. 4b).

Our quantitative parallel reaction monitoring (PRM) analysis confirmed the significant activation of γ-H2A.X (H2A.X-S139) after

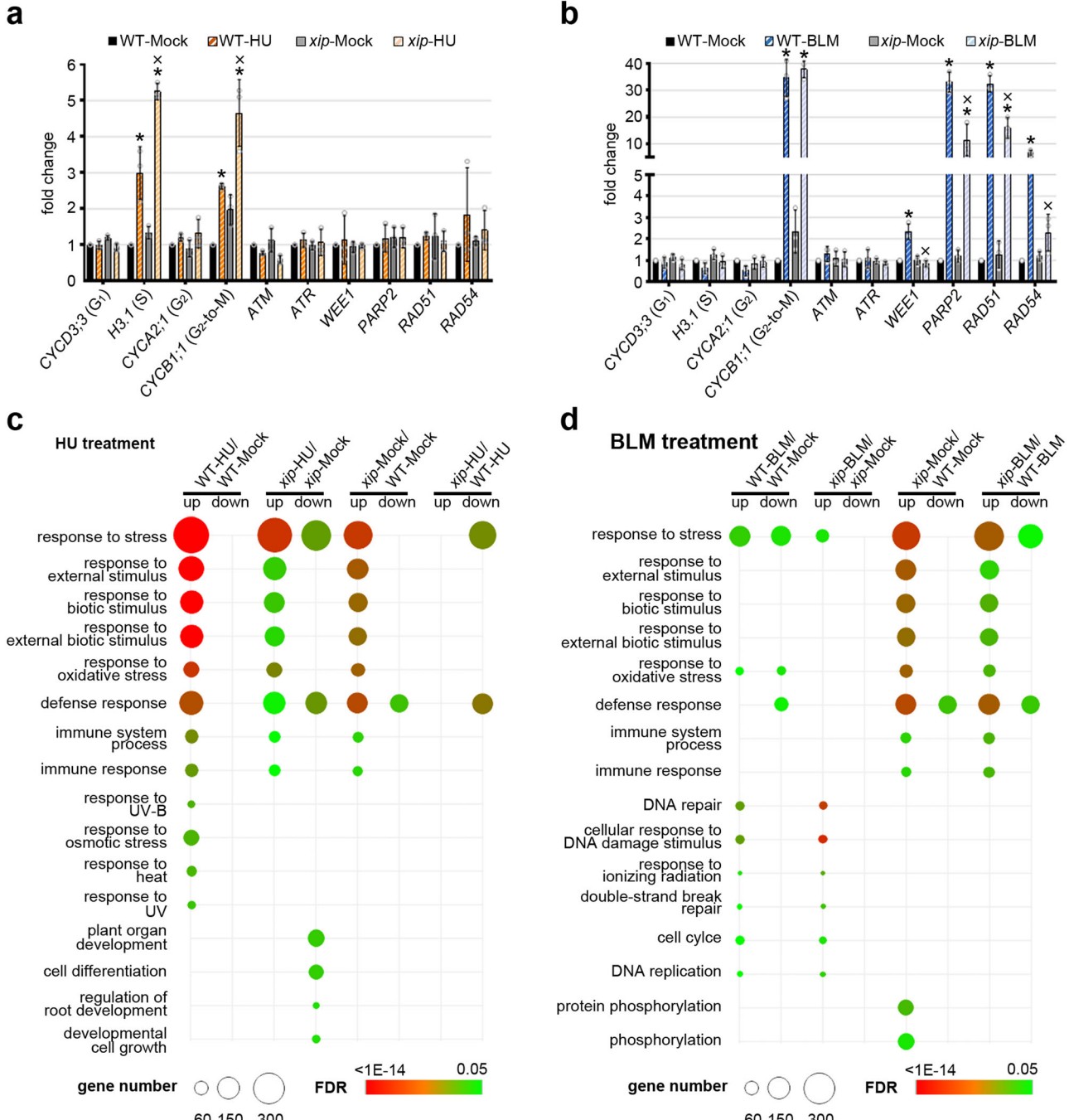

**Fig. 3 | The transcriptional changes in *xip* mutant. a** RT-qPCR analysis of phase-specific cell cycle genes (the corresponding phase is shown in parentheses) and DNA repair-related genes in 12-day-old WT and *xip* seedlings grown with or without 24-hour-long treatment of 1 mM HU. *ACT2* was used as reference gene. Mean values of the relative transcript levels in the analyzed samples compared with those in WT under mock condition (WT-Mock, set as 1) were shown together with error bars indicating ±SD from three independent biological replicates. Asterisks indicate both statistically significant difference (*p* < 0.05, *t*-test, two-tail) and fold change > 2 in samples when compared with WT-Mock. "X" indicates statistically significant difference (*p* < 0.05, *t*-test, two-tail) in *xip*-HU when compared with WT-HU sample. All the *p* values can be found in the source data. **b** RT-qPCR analysis of cell cycle and DNA repair-related genes in 12-day-old WT and *xip* seedlings grown with or without 6-hour-long treatment of 2.5 µM BLM. *ACT2* was used as reference gene. Mean

values of the relative transcript levels in the analyzed samples compared with those in WT under mock condition (WT-Mock, set as 1) were shown together with error bars indicating ±SD from three independent biological replicates. Asterisks indicate both statistically significant difference (*p* < 0.05, *t*-test, two-tail) and fold change > 2 in samples when compared with WT-Mock. "X" indicates statistically significant difference (*p* < 0.05, *t*-test, two-tail) in *xip*-BLM when compared with WT-BLM sample. All the *p* values can be found in the source data. **c** GO analysis of significantly up- or downregulated genes in four gene sets, WT-HU/WT-Mock, *xip*-HU/*xip*-Mock, *xip*-Mock/WT-Mock and *xip*-HU/WT-HU. The size of each circle represented gene number, and the false discovery rate (FDR) values were indicated by various colors. **d** GO analysis of significantly up- or downregulated genes in four gene sets, WT-BLM/WT-Mock, *xip*-BLM/*xip*-Mock, *xip*-Mock/WT-Mock and *xip*-BLM/WT-BLM.

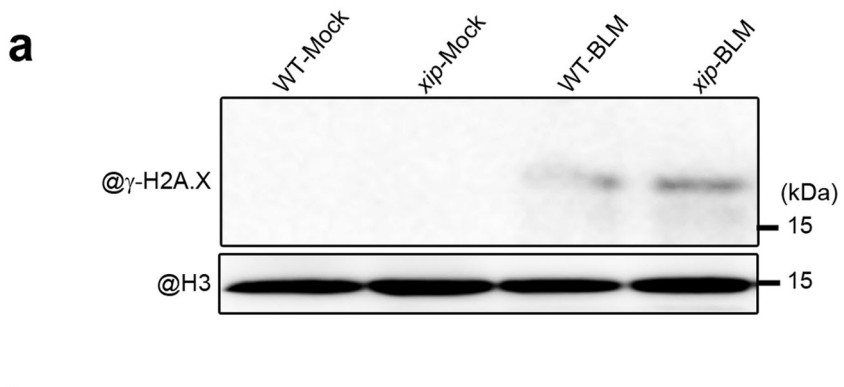

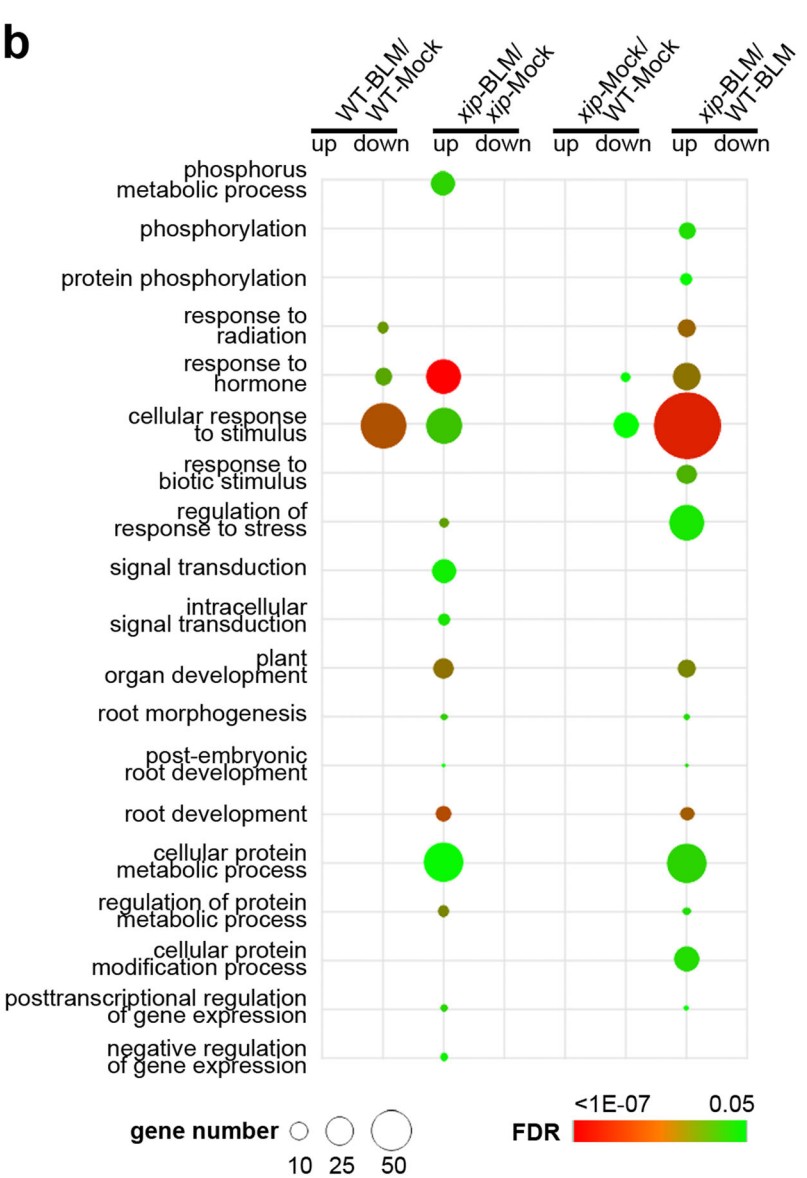

**Fig. 4 | The changed phospho-proteomics in *xip* mutant. a** Comparison of γ-H2A.X in different samples. 12-day-old seedlings were treated with or without 2.5 μM BLM for 6 hours, and used for histone/protein extraction. H3 antibody was used as a loading control. The antibody against γ-H2A.X was generated in our previous study[20]. Representative results from three independent Western blot experiments were shown. **b** GO analysis of significantly up- or down-regulated DPPs in four protein sets, WT-BLM/WT-Mock, *xip*-BLM/*xip*-Mock, *xip*-Mock/WT-Mock and *xip*-BLM/WT-BLM. The size of each circle represented gene number, and the false discovery rate (FDR) values were indicated by various colors.

BLM treatment (Supplementary Fig. 13 and Data 6). Besides, Ser or Thr residue within proteins implicated in cell cycle progression and cell death were found significantly hyper-phosphorylated in *xip* mutant compared to WT after BLM treatment (Supplementary Fig. 13 and Supplementary Data 6). Taken together, our phospho-proteomic analysis provided insight into the mis-regulated plant DNA damage signaling pathway at the layer of protein phosphorylation.

## Chromatin instability and DNA damage in *xip* mutant

We then examined the genome integrity by using single-cell gel electrophoresis assay, an assay generally also known as comet assay[26]. We found that, even under mock growth condition, there were slightly more DNA fragments with significant difference in *xip* mutants when compared to WT. The difference was much more obvious and significant after BLM treatment (Supplementary Fig. 14a).

Propidium Iodide (PI) cannot enter the cell wall of living plant cells and merely mark dead cells because of their interrupted membrane integrity. We found that the PI-stained dead cells could be found in WT root tips only after BLM treatment. In comparison, the root apical meristem of *xip* mutants showed sporadic cell death even under mock growth condition, and more PI-stained cell death was found under BLM treatment (Supplementary Figs. 14b and 15), indicating that *XIP* is required to impede spontaneous and DSB-induced cell death in vivo. Complete *XIP* gene sequence (*XIP* in *xip*) complemented the phenotype of *xip* mutants in plant cell death and genomic instability (Supplementary Fig. 14a, b), indicating the lose-of-function of *XIP* was the cause of these phenotypes.

The plant-specific transcription factor SOG1 has been proved as a master factor to govern the downstream DDR, including the transcriptional induction of DDR genes and DSB-induced cell death[27]. As previously reported[28], in great contrast to WT, *SOG1*-depleting mutant *sog1-101* showed no cell death and no transcriptional induction of DDR genes such as *RAD51* and *RAD54* in response to BLM treatment (Fig. 5a, b). Although there is no evidence that distinct ecotypes, such as Ws ecotype (*xip* in this study) and Col ecotype (*sog1-101*), have a clear difference in Arabidopsis DDR pathway, we crossed *xip* with WT [Col] for four consecutive generations, which is referred as *xip* [Col] below, to minimize the influence of different backgrounds on subsequent observations. Similarly, the *xip* [Col] also showed more cell death phenotype at both mock and BLM conditions when compared to WT [Col] (Fig. 5a and Supplementary Fig. 15). We then introgressed this mutant into *sog1-101* mutant[28]. Intriguingly, the double mutant *sog1-101 xip* [Col] showed similar cell death and transcriptional response with *sog1-101* (Fig. 5a, b), indicating that these phenotypes implicated in DDR pathway caused by XIP depletion rely on the SOG1 activity.

## XIP interacts with RAD51 and somatic homologous recombination is reduced in *xip* mutant

As a reader protein of DSB-labeling γ-H2A.X, XIP needs to transfer the DSB signals to downstream DNA repair machinery, which is proposed to require further protein-protein interplay. Therefore, we surveyed the XIP-interacting proteins (XIPIPs) through pulldown-MS analysis. In this study, 15 XIPIPs were finally identified. RAD51, the key recombinase involved in homologous recombination (HR) pathway in DSB repair[29], and another BRCT-like domain-containing protein encoded by AT2G41450, were included in the interacting protein list (Supplementary Data 7). Notably, γ-H2A.X, which cannot be solubilized by the neutral extraction buffer[30], was not identified in this assay.

We verified the interaction of XIP with RAD51 by pulldown experiment. GST-fused RAD51, but not GST, specifically interacted with full-length XIP-M (Fig. 5c). We also expressed N-terminally Myc-tagged XIP-M and co-expressed it with previously reported FLAG-tagged RAD51[31] in mesophyll protoplasts. In the co-immunoprecipitation (co-IP) experiment, RAD51-FLAG was specifically detected in the IP fraction of Myc-XIP-M (Fig. 5d). We also verified their interaction via bimolecular fluorescence complementation (BiFC) experiment (Supplementary Fig. 16). Taken together, these results indicated that XIP and RAD51 can form protein complex in vivo.

Considering the crucial role of RAD51 in HR, we speculated that the interaction of XIP with RAD51 may facilitate efficient DSB repair through HR pathway. To test this hypothesis, we examined the somatic HR frequency (HRF) in seedlings in response to BLM treatment

through the application of model recombination substrates. *1445* (Fig. 5e) and *IC9C* (Fig. 5f) were used to measure the intramolecular and intermolecular HRF in Arabidopsis, respectively[32]. The somatic HR events recombined the incomplete *GUS* fragments into complete *GUS* gene through corresponding mechanisms, which were represented as GUS spots in plants after staining. The HRF of both *1445* and *IC9C* in WT [Col] obviously increased with the increase of BLM concentration, indicative of the gradual activation of HR pathway in DSB repair. However, HRF in *xip* [Col] mutant was significantly inhibited under all the examined conditions when compared to WT, indicating that XIP activity was important for full HR activation in plants. This finding of decreased HRF was also consistent with the increased chromatin instability found in *xip* mutant. It may also be possible that there is a difference in somatic mutagenesis.

## Discussion

The sensing and signaling of DNA damage, and the consequent coordination with DNA repair process, all occur in the context of chromatin structure. Albeit critical to all the organisms, these pathways have been less characterized in plants due to the missing of many conserved players. In this study, we identified a plant γ-H2A.X-interacting protein, XIP, the major isoforms of which comprise of dual-BRCT-like domain (RTT107_BRCT_5 followed by BRCT). Detailed investigation was performed on the biological function of *XIP* in plant growth and resistance to genotoxic stress. *XIP* is required for the full activation of multiple DDR genes in face of genotoxin challenge. We found *XIP* function is highly related to the DNA damage-related transcriptome and phospho-proteome in plant. While the precise molecular function of XIP remains to be determined, based on the phenotype and the protein data showing XIP can interact with γ-H2A.X and RAD51, we propose a possible model for how XIP could contribute to HR and the maintenance of genome stability (Supplementary Fig. 17).

Although playing a vital role in plant DDR pathway, the phosphorylation cascade triggered by ATM and ATR displays little substrate specificity for serine or threonine residues, thus resulting in the necessity for various binding modules to distinguish their specific targets[24]. In consistence, BRCT-like domain(s), as a phospho-protein-binding domain, has evolved into several families with delicate sequence and/or structure variations and thereinafter potential functional divergence. In addition, more complexity has been introduced by the architecture of BRCT-like domains, which can be individual, or serial, or in assembly with other functional domains, such as the distinct arrangement found in BRCA1, ROW1 and LIG4 proteins. Such diversity may provide necessary specificity of BRCT-like domains in the recognition with their targets[15].

The protein encoded by the shortest transcript of *XIP*, NM_001340430.1, comprises only a single-BRCT-like domain. Compared to the dual-BRCT-like domain, the binding interface between a single-BRCT-like domain and its putative cognate phosphorylated peptide was proposed to be less extensive, resulting in weakened recognition specificity and/or binding strength. The in vitro ppd assay showed that single-BRCT-like domain in XIP-S did not effectively retrieve p2/p3 peptides. Although the possibility of weak interaction under less stringent elution conditions cannot be ruled out, we speculate that such single-BRCT-like domain may require additional functional domains or proteins to form oligomeric protein complex to strengthen or stabilize such recognition. Another hypothesis is that the single-BRCT-like domain may be involved in a more dynamic signal transduction process. For instance, single-BRCT-like domain can interact with another independent single domain through an interface distinct from the hydrophobic dual-BRCT-like domain. Evidence has been provided that independent BRCT domains can form homo- or hetero-oligomers in vitro[33]. Although less characterized, the interaction of independent single-BRCT-like domains has been found in XRCC1 and PARP1, which negatively regulates PARP1 activity[33,34].

 

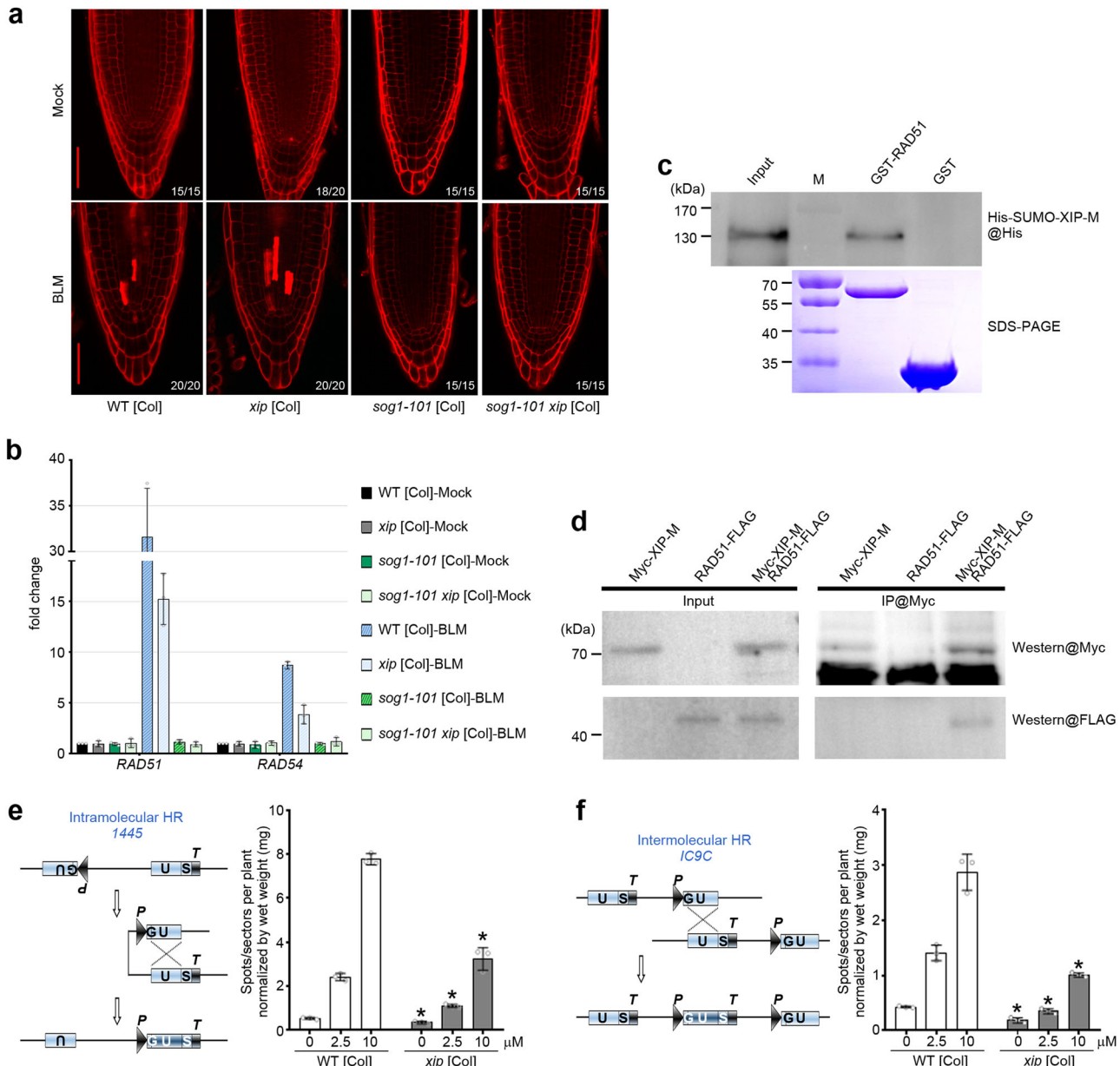

**Fig. 5 | XIP interacts with RAD51 and HR is reduced in *xip* mutant.**
**a** Representative image of 15-20 PI-stained root tips of 6-days-old WT [Col], *xip* [Col], *sog1-101* [Col], and *sog1-101 xip* [Col] seedlings with or without 6-hour-long treatment of 2.5 μM BLM. Bar = 50 μm. **b** RT-qPCR analysis of *RAD51* and *RAD54* genes in 12-day-old seedlings (all in [Col]) grown with or without 6-hour-long treatment of 2.5 μM BLM. Mean values of the relative transcript levels in the analyzed samples compared with those in WT [Col] under mock condition (set as 1) were shown together with error bars indicating ±SD from three independent biological replicates. **c** RAD51 was expressed as GST-fused protein (lower panel) and used in pulldown assay (upper panel). His-SUMO-tagged full-length XIP-M was detected by anti-His antibody in pulldown fraction of GST-RAD51 but not GST control. Representative result from two independent pulldown experiments was shown. **d** co-IP detection of Myc-XIP-M and RAD51-FLAG interaction *in planta*. Total protein extracts were first immuno-precipitated with anti-Myc antibody, and the resulting fractions were then analyzed in Western blot by using anti-Myc (upper panel) or anti-FLAG antibody (lower panel). Representative result from two independent co-IP experiments was shown. **e** Scheme of GUS-based recombination

substrate and intramolecular HR in line *1445*. Intramolecular interaction between inverted repeats (U) results in the restoration of a functional *GUS* gene through intramolecular HR (left panel). Spots/sectors per plant (SPP) were assessed for WT [Col] and *xip* [Col] on a population of approximately 50 individual plants germinated and grown with or without indicated concentration of BLM (right panel) in one biological experiment. HRF assessment has been performed in three independent experiments. Mean values were shown together with error bars indicating ±SD. Asterisks indicated the statistically significant differences between WT and *xip* mutant (*p* < 0.05, *t*-test, two-tail). All the *p* values can be found in the source data. **f** The HR frequency assessment by intermolecular HR marker *IC9C*. Spots/sectors per plant (SPP) were assessed for WT [Col] and *xip* [Col] on a population of approximately 50 individual plants germinated and grown with or without indicated concentration of BLM (right panel) in one biological experiment. HRF assessment has been performed in three independent experiments. Mean values were shown together with error bars indicating ±SD. Asterisks indicated the statistically significant differences between WT and *xip* mutant (*p* < 0.05, *t*-test, two-tail). All the *p* values can be found in the source data.

However, due to the low abundance of this *XIP* transcript isoform, whether XIP-S participates in other signaling processes independent of γ-H2A.X remains unclear. It is not easy to distinguish the relative contribution of these isoforms so far.

In this study, we showed that such dual-BRCT-like domain, but not single-BRCT-like domain, can specifically interact with γ-H2A.X. We proposed that, complementary to the phosphorylation cascade driven by diverse kinases, distinct phospho-readers, including diverse BRCT-

like domain-containing proteins, play a critical role to guarantee the orderly transmission of complicate signals in the whole DDR pathway.

The huge difference in N-terminal moiety of XIP and MDC1 raised a possibility that dual-BRCT-like domain is solely responsible for γ-H2A.X binding, while the remaining domain in these readers may be implicated in recruiting species-specific cognate repair proteins. One evidence came from the studies in yeast. There are at least two γ-H2A.X-interacting proteins in *Schizosaccharomyces pombe*, Brc1, and Crb2, mediating the γ-H2A.X signal in response to replicative and genotoxic stresses, respectively[35]. Both two readers use their highly conserved C-terminal BRCT-like domains to recognize γ-H2A.X, however, with difference in their other domains which may be involved in recruiting distinct repair proteins. In an attempt to survey the interacting proteins of XIP, we identified RAD51 protein, the key recombinase involved in HR, which connects XIP activity with DNA damage repair.

Increasing evidence has been provided that human and murine MDC1 harbor many protein-binding modules and mainly act as a multifaceted mediator recruiting DDR proteins to DSB sites[16,36]. MDC1 physically interacts with ATM via its N-terminal Forkhead-Associated (FHA) domain, which facilitates the recruitment of ATM to the DSB sites and its chromatin retention, forming a positive feedback loop for γ-H2A.X foci formation[17,18]. The formation of γ-H2A.X foci was significantly impaired in cells depleting of MDC1 protein, indicative of an important role of MDC1 in the establishment of DSB-induced γ-H2A.X[37]. However, the plant γ-H2A.X intensity was instead increased in *xip* mutant, suggesting that XIP is dispensable for the ATM-dependent γ-H2A.X formation and subsequent amplification. Therefore, XIP is required for the maintenance of chromatin stability to avoid more DNA breaks and consequent γ-H2A.X accumulation.

In addition to ATM, FHA domain of MDC1 also mediates its interaction with animal-specific Chk2, a central effector kinase responsible for cell cycle checkpoints and apoptosis[38] and DNA repair protein such as RAD51[39]. MDC1 can also directly interact with animal-specific 53BP1 via its BRCT domain[40]. Moreover, MDC1 itself can be phosphorylated in many motifs and thus be recognized by component in MRN complex and animal-specific RNF8[16]. However, apart from the BRCT-like domain(s), Arabidopsis XIP has nearly no significant similarity with mammal MDC1 in protein sequence, and our phospho-proteomic analysis also provided evidence that no phosphorylated peptide of XIP has been found in this study (Supplementary Data 3). Notably, we found a similar protein complex of RAD51-MDC1 in mammal and RAD51-XIP in Arabidopsis, which can both connect the role of MDC1/XIP with DNA damage repair via HR. However, RAD51 forms a complex with MDC1 through a direct interaction with the FHA domain of MDC1, but not its BRCT-like domain, which excluded a possibility of conserved interaction interface in these protein complexes. Moreover, the FHA domain is also indispensable for MDC1-mediated HR[39]. Although lacking of the necessary FHA domain, our pulldown assay revealed that XIP can directly interact with RAD51, implying the existence of a different protein-protein interaction interface, likely resulted from the tertiary structure through protein folding, which requires the elucidation by future crystal analysis of plant XIP proteins. However, since HRF is only downregulated, rather than eliminated, in *xip* mutant, it is quite reasonable that there may exist other mechanisms to recruit RAD51 to DSB site, which accommodates the urgent requirement of DNA repair machinery in response to DSB stress. Actually, Arabidopsis retinoblastoma protein (Rb) homolog RBR1, a transcriptional regulator, has been found to be required for RAD51 localization to DNA lesions in a CDKB1 activity-dependent manner[41]. In human cells, BRCA1-BARD1 complex was reported to interact with Partner and Localizer of BRCA2 (PALB2), which further recruited RAD51 to the proximity of DSB site[42]. Although PALB2 has no plant homologs[43], whether or not plant BRCA1-BARD1 homolog can recruit RAD51, or if

true, through an alternative and plant-specific manner, still remains further analysis to elucidate.

## Methods

### Plant materials and growth conditions

In vitro plant culture was performed on agar-solidified Murashige and Skoog (MS) medium supplemented with 0.9% sucrose. Plants were cultivated at 21 °C under a 16-h-light/8-h-dark photoperiod in a Percival AR41L5 growth chamber in 60 to 90 μE m$^{-2}$ s$^{-1}$ white light (LED lamp). For plant phenotype observation, seedlings were germinated and grown on medium containing 1 mM HU or 2.5 μM BLM for indicated days, respectively. The T-DNA insertion mutant *xip* (FLAG_589H07) was derived from the Wassileskija ecotype. The *sog1-101* mutant and the HR reporter lines *1445* and *IC9C* in Col ecotype have been previously studied[28,32]. We crossed *xip* with WT [Col] for four consecutive generations to introgress *xip* mutation into Col background (referred as *xip* [Col]).

### Vector construction

The full-length *XIP* genomic DNA (including 2 kb of promoter and 1 kb of terminator of the longest *XIP-L*) was cloned into modified pCAMBIA1301 vector without *35S* promoter and YFP-encoding sequence, and was transformed into *Agrobacterium tumefaciens* strain GV3101. The resulting strain was used to transform Arabidopsis using the floral dip method.

For the sub-cellular localization and immunostaining assays, the CDS of XIP-M was cloned into *Eco*RI-cut pCAMBIA1301 vector to express YFP-XIP-M. For the expression of recombinant proteins in bacterium, DNA fragment encoding 6×His-SUMO-tagged XIP-M was cloned into *Eco*RI/*Sal*I-digested pET-28a (Novagen), and the DNA fragment encoding single/dual-BRCT-like domains and the CDS of RAD51 were cloned into *Bam*HI/*Sal*I-digested and *Eco*RI-digested pGEX-6P-1, respectively, to express GST-tagged proteins.

For co-IP experiment, the CDS of XIP-M was cloned into *Sac*I/ *Spe*I-digested pRTVnMyc vector to express Myc-XIP-M, and the CDS of RAD51 was cloned into *Bam*HI-digested pCAMBIA1306 to express RAD51-FLAG.

### Rapid Amplification of cDNA End (RACE) assay

The 5'- and 3'-RACE assays were performed following the instructions of SMARTer 5'-RACE and 3'-RACE (Clontech, Mountain View, CA, USA; Code No. 634858 and 634859). The cloned RACE fragments were aligned to find the intact 3'-terminus of the longest transcript of *XIP* in WT and the 5'-terminus of abnormal transcript in *xip* mutant, respectively.

### Bimolecular Fluorescence Complementation (BiFC) assay

The CDS of *RAD51* and *XIP-M* were cloned into pXY105 (Yc) and pXY106 (Yn) vectors[44], respectively, for transgene constructions in BiFC assay. Leaves of *Nicotiana benthamiana* plants were co-infiltrated with *Agrobacterium tumefaciens* strain GV1301 carrying transgene constructs and/or empty vectors (negative controls). BiFC fluorescence was observed at 2–3 days after infiltration by using a confocal laser scanning microscope (LSM 710; Zeiss, Oberkochen, Germany).

### peptide pulldown (ppd)-MS experiment

N-terminally biotinylated peptides were synthesized by Scilight-peptide (http://www.scilight-peptide.com/). Peptides were coupled to Dynabeads™ M-280 Streptavidin (ThermoFisher, Cat: 11205D) in PBST buffer (PBS, pH 7.4, containing 0.01% [v/v] Tween™−20) at the concentration of 200 pmol of peptide/1 mg of beads. The beads were incubated with Arabidopsis protein extracts for ppd-MS, or incubated with recombinant GST-fused BRCT-like domains or recombinant His-SUMO-XIP-M protein for ppd verification. Beads were extensively washed with PBST buffer and the bound proteins were subjected to

SDS-PAGE, which was followed by mass spectrometry. Briefly, the trypsin-digested peptides were sprayed into an LTQ-Orbitrap Elite mass spectrometer (Thermo Fisher Scientific, Waltham, MA, USA) equipped with a nano-electrospray ionization ion source. Alternatively, the proteins were detected by silver staining[45].

## Arabidopsis mesophyll protoplast and microscopy

The preparation of mesophyll protoplast and the transient gene expression were carried out following the published protocol[46]. We performed the immunostaining and imaging of YFP-XIP-M and γ-H2A.X in protoplasts[25]. Briefly, protoplasts were cultured at room temperature for 20 hours for protein expression, and at the last 3 hours, BLM was added to a final concentration of 2.5 µM to induce γ-H2A.X. After that, protoplasts were collected and fixed for at least 3 hours using 4% PFA/PBS. After extensive washes by PBS at room temperature, protoplasts were mounted on slides covered with poly-lysine. Anti-γ-H2A.X (rabbit)[25] and anti-GFP (mouse) (Abmart, China; Code No, 20004 L) were added at 1:200 dilution ratio. Fluorescent secondary antibodies (Invitrogen, Code No, A11001, A21428) was added at 1:1000 dilution ratio. Confocal images were acquired by using a LSM880 microscope (Zeiss).

Differential interference contrast (DIC) images were taken with an Imager A2 microscope (Zeiss)[47]. For PI staining, all the root tips were transferred to liquid medium with or without 2.5 µM BLM for 6 hours. Confocal images were acquired by using a LSM880 microscope (Zeiss). At least 15 root tips per sample were used in these observations.

## Transcript and Transcriptome analysis

12-day-old seedlings were transferred to liquid medium with or without 1 mM HU for 24 hours (HU and mock samples). In the case of BLM, seedlings were first transferred to normal liquid medium for 18 hours and then transferred to liquid medium containing 2.5 µM BLM for additional 6 hours (BLM samples). RNA was extracted from seedlings by using TRIzol kit according to standard procedure (Invitrogen, No, 10296010). Reverse transcription (RT) was performed using Improm-II reverse transcriptase (Promega, Madison, WI, USA). RT-qPCR was performed in three biological replicates according to the standard instruction[48]. The primers used in this study were listed in Supplementary Data 8. Two replicates of WT and *xip* RNA were subjected to RNA-seq analysis and analyzed under standard procedure[49]. Briefly, the RNA-seq libraries were constructed by following the KAPA stranded mRNA-seq Kit instructions for Illumina® Platforms (Kapa Biosystems, Code No, KR0960-v5.17), and sequenced on an Illumina HiSeq3000 instrument via the custom service of GENERGY BIO (Shanghai, China). DESeq2 v1.16.1 software was used to screen differentially expressed genes (DEGs) between different sample groups. The significant up- and downregulated DEG genes were screened under a stringent criterion: both fold change ≥ 2 and *p*-value ≤ 0.05 (adjusted with the Benjamini-Hochberg correction). Volcano map production was done using the R language ggplot2 package[50]. DEGs enrichment analysis was done using agriGO v2.0 (http://systemsbiology.cau.edu.cn/agriGOv2/)[51].

## Histone extraction and Western blot

12-day-old seedlings were transferred to liquid medium with or without 2.5 µM BLM for 6 hours (BLM and mock samples) for histone/protein extraction. Histones were extracted from indicated seedling samples according to the standard acid extraction protocol[32]. The plant-specific γ-H2A.X antibody was prepared by Abmart (http://www.abmart.com.cn) in our previous study[25]. Anti-H3 antibody (Abcam, ab1791) was used for loading control.

## Sample digestion and phospho-peptide enrichment

The protein extraction and following FASP digestion were adapted for the following procedures in Microcon PL-10 filters[52]. After three successive rounds of buffer displacement with 8 M Urea and 100 mM Tris-HCl (pH 8.5), the proteins were reduced by 10 mM DTT at 37 °C for 30 min, and followed by alkylation with 30 mM iodoacetamide at 25 °C for 45 min in dark. Then the sample was displaced with digestion buffer (30 mM Tris-HCl, pH 8.0) for three times. The digestion was carried out with trypsin (enzyme/protein as 1:50) at 37 °C for 12 hours. After digestion, the solution was filtrated out and the filter was washed twice with 15% acetonitrile (ACN), and all the filtrates were pooled and vacuum-dried. The home-made $TiO_2$ microcolumns (peptide: $TiO_2$ = 1:10) were washed by 100% ACN. Peptides were dissolved with loading buffer (1 M glycolic acid, 80% ACN and 5% trifluoracetic acid (TFA)). After loading peptides onto the microcolumn twice, the microcolumn were washed twice with loading buffer and then with washing buffer (80% ACN and 1% TFA) twice. The phosphopeptides were successively eluted by 2 M ammonia hydroxide and then 1 M ammonia hydroxide with 30% ACN. The eluted samples were vaccum-dried by SpeedVac.

## LC-MS analysis and data processing

NanoLC-MS/MS analysis was performed using an EASY-nLC 1200 system (Thermo Fisher Scientific, USA) coupled to an Orbitrap Fusion Lumos mass spectrometer (Thermo Fisher Scientific, USA). A one-column system was adopted for all analyses. Samples were analyzed on a home-made C18 analytical column (75 µm i.d. × 25 cm, ReproSil-Pur 120 C18-AQ, 1.9 µm (Dr. Maisch GmbH, Germany))[53]. The mobile phases consisted of Solution A (0.1% formic acid) and Solution B (0.1% formic acid in 80% ACN). The peptides for LFQ analysis were eluted using the following gradients: 2–5% Solution B in 2 min, 5–35% Solution B in 100 min, 35-44% Solution B in 6 min, 44–100% Solution B in 2 min, 100% Solution B for 10 min, at a flow rate of 200 nL/min. The phospho-peptides were eluted using the following gradients: 2-5% Solution B in 3 min, 5-35% Solution B in 40 min, 35-44% Solution B in 5 min, 44-100% Solution B in 2 min, 100% Solution B for 10 min, at a flow rate of 200 nL/ min. Data acquisition mode was set to obtain one MS scan followed by HCD-MS/MS acquisitions with a cycle time of 2 seconds. The normalized collision energy (NCE) was set as 30. The results were processed with Arabidopsis database TAIR (32790 entries, 201307) and the using Protein Discoverer (version 2.4, Thermo Fisher Scientific) with Mascot (version 2.7.0, Matrix Science)[54]. The ptmRS node was used in the workflow[55]. The mass tolerances were 10 ppm for precursor and fragment Mass Tolerance 0.05 Da. Up to two missed cleavages were allowed. The carbamidomethylation on cysteine as a fixed modification, and acetylation on the protein N-terminal and oxidation on methionine as variable modifications. For phospho-peptides, additional parameters were set: the phosphorylation on serine, threonine and Tyrosine as variable modifications.

## Parallel reaction monitoring (PRM)

The sample prepare procedures were performed as previous description. Briefly, after trypsin digestion, the separated peptides were analyzed in Orbitrap Fusion Lumos with an NSI ion source. The electrospray voltage applied was 2.0 kV. The full MS scan resolution was set to 60,000 for a scan range of 350–1400 m/Z, and the further MS/MS scan resolution was set to 30,000. The automatic gain control (AGC) was set at 4E5 for full MS and 5E4 for MS/MS, the maximum IT was set at 54 ms, the isolation window for MS/MS was set at 1.6 m/z.

The MS data were processed with SpectroDvie (10.1). The enzyme was set as trypsin/P and the maximum miss cleavages was 2. The carbamidomethylation on cysteine as a fixed modification, and acetylation on the protein N-terminal and oxidation on methionine as variable modifications. For phospho-peptides, the phosphorylation on Serine, Threonine and Tyrosine were set as variable modifications. The peptide length was set as 7–52, and the maximum number of variable

modifications was set as 5. The precursor charges were set as 2–4, the fragment m/z were set as 300–1800, the maximum fragment charge was set as 3, and ion types were set as b and y. $H_2O$, $NH_3$, and no loss were allowed. The top 6 fragments were used for quantification and the minimum fragment length were set as 3.

## Pulldown and co-IP

The pulldown experiment was performed following a standard protocol[56]. Briefly, immobilized GST-RAD51 and GST (control) on glutathione-Sepharose 4B beads were mixed with His-SUMO-XIP-M for 4 hours at 4 °C in binding buffer (1 × PBS, 0.2% NP-40, 1 mM DTT and protease inhibitor cocktail (Roche)), followed by wash buffer (20 mM Tris-HCl (pH8.0), 0.5 M NaCl, 0.5% NP-40, 1 mM DTT and protease inhibitor cocktail) for 3 times. The proteins remained on the beads were resolved by SDS-PAGE, and examined in Western blot by using anti-His antibody (Shanghai Genomics, SG4110-05).

The co-IP experiment was performed following a standard protocol[49]. Briefly, Myc-XIP-M and RAD51-FLAG were individually expressed or co-expressed in Arabidopsis protoplasts. These proto-plasts were collected and lysed in lysis buffer (50 mM Tris–HCl, pH 7.5, 150 mM NaCl, 5 mM $MgCl_2$, 10% glycerol, 0.1% NP-40, 5 mM DTT, 1 mM PMSF, and proteinase inhibitor cocktail (Roche)) to extract soluble proteins. The extract was immuno-precipitated by anti-Myc antibody (Abmart, China; Code No, M20002L, 1:5000 dilution) overnight at 4 °C and magnetic protein A beads (Invitrogen, 10002D) were added for 2 hours for protein enrichment. The immuno-precipitates were washed four times with wash buffer (2 mM $KH_2PO_4$, 10 mM $Na_2HPO_4$, 140 mM NaCl, 2.7 mM KCl, 0.05% (v/v) Tween 20), and analyzed by Western immunoblot using anti-Myc and anti-FLAG (Sigma-Aldrich, F1084). Signal was captured using Chemiluminescence Imaging System (ChemiScope 3600 Mini, ClINX).

## Comet assay and HRF assessment

12-day-old seedlings were transferred to liquid medium with or without BLM for 6 hours of treatment, and were collected for comet assay[57]. Briefly, seedlings were sliced in PBS containing 50 μM EDTA with a fresh razor blade to release nuclei. Released nuclei were mounted and fixed onto microscopic slides pre-coated with agarose, and were subjected to lysis in high salt buffer (2.5 M NaCl, 10 mM Tris-HCl, pH7.5, 100 mM EDTA) for 20 min. The slides were equilibrated in TBE buffer and electrophoresed in the same buffer. 10 μg/ml of PI was used for DNA/nuclei staining. HRF was assessed by histochemical GUS staining and observation using a stereomicroscope[32]. The number of HR events (approximately 50 plants per sample) was assessed visually using a stereomicroscope. HRF assessment has been performed in three independent experiments.

## Accession numbers

Sequence data from this article can be found in the GenBank under the following accession numbers: XIP, AT4G03130; CYCD3;3, AT3G50070; H3.1, AT1G09200; CYCA2;1, AT5G25380; CYCB1;1, AT4G37490; ATM, AT3G48190; ATR, AT5G40820; WEE1, AT1G02970; PARP2, AT4G02390; RAD51, AT5G20850; RAD54, AT3G19210; BSK1, AT4G35230; FBR12, AT1G26630; H2A.X, AT1G08880. The T-DNA insertion mutant *xip* (FLAG_589H07) was obtained from the ABRC.

## Reporting summary

Further information on research design is available in the Nature Portfolio Reporting Summary linked to this article.

## Data availability

Data supporting the findings of this work are available within the paper and its Supplementary Information files. The raw and processed data of RNA-seq have been deposited in NCBI-GEO (GSE196370). The phosphoproteomics and proteomics data have been deposited in ProteomeXchange (PXD037072). Source data are provided with this paper.

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

## Acknowledgements

This research work was financially supported by the National Natural Science Foundation of China to Yan Zhu (Grant NSFC 32070201 and 32270629) and to Huijia Kang (Grant NSFC 32200465).

## Author contributions

Y.Z. conceived and designed the research. Y.Z. supervised the experiments. T.F., H.K., D.W., X.Z., L.H., J.W. performed the experiments. Y.Z. wrote the manuscript. All authors read, revised, and approved the manuscript.

## Competing interests

The authors declare no competing interests.
