## [Peer Review File · Nature Communications]

Arabidopsis γ -H2A.X-INTERACTING PROTEIN participates in DNA damage response and safeguards chromatin stabilityReviewer #1 (Remarks to the Author):

The editor asked me to look at the quality and reliability phospho-proteomics experiments. Therefore, I have only taken a closer look at the phospho-proteomics data here.

At first I would like to give the authors here some well-meant advice. In my opinion, it would be a great idea to present the results clearly and not to overwhelm the reviewers with excel tables where it is not clear what has been done. Up to now, the technical presentation of the data has been inadequate or, for me, incomprehensible. However, a reviewer should be able to understand all the processes in the calculation of the data.

Comments:

Major item:

Phospho-proteomics: all relevant phospho-peptides and the corresponding non-phosphorylated peptides should be presented in a separate excel table.

Precursor information from Protein Discoverer search result is not very good quantitative info. It is common today that peak areas are calculated. Please calculate these peak areas with XCalibur or a quantitative software which is integrated in Protein Discoverer (e.g. qp-quant). The peak areas should be copied and displayed in a supplemental figure.

The calculation of the phospho-peptide to non-phospho-peptide ratio should be comprehensible and the standard deviation of the peak areas should be calculated.

Please note that the sequence coverage of some proteins of interest is low (< 15 %). Therefore, it's nearly impossible to find all potential modifications at the proteins of interest (γ -H2A.X, H2A.W.7,...).

Anyhow, to get a better picture of quantitative data interpretation, I opened 3 data files with Xcalibur (blm-wt, blm-xip, mock) and extracted the mass of a phospho-peptide (602.2675). Based on extracted ion chromatograms, I can see a slight upregulation (factor 2) for blm-wt. Therefore I cannot confirm the statement of the authors ("S3 and S131 phosphorylation in H2A.W.7, which mediated DNA damage response in heterochromatin21, were also found significantly increased in xip mutant compared to WT after BLM treatment").

Conclusion: From my side, I cannot accept this publication as it is.

To obtain more accurate or better quantification information, a re-analysis of the data with targeted proteomics (PRM) is necessary, absolutely.

Reviewer #2 (Remarks to the Author):

In eukaryotes, induction of a DNA DSB promotes the phosphorylation of H2AX (by localized, activated ATM) near the break to produce a zone of gamma-H2AX (gH2AX). This phosphorylation is thought to act as a positive feedback loop, further activating ATM and so enhancing the DNA Damage Response, until the break is repaired and the phosphorylated H2AX removed. Plants have been shown to gamma-phosphorylate H2AX, though the significance of this event has, surprisingly, not been investigated. This paper focuses on the discovery and characterization of mutants defective in a protein that the authors describe as a "reader" for gH2AX. The protein was identified as binding to a peptide matching the phosphorylated C-terminus of H2AX, but not to the peptide's unphosphorylated version.

The term "reader" implies, to me, that it is an adaptor between gH2AX and the DNA damage response, and therefore required for gH2AX's role in DDR. To determine the role of XIP in response, the authors characterize a mutant defective in this gene (named XIP). They find that some aspects of the DDR are expressed constitutively (at a mild level, in plants not intentionally exposed to exogenous DNA damaging agents), transcriptional response to DSB inducing agents still present, though perhaps muted, and some aspects of DDR may be mildly enhanced (such as premature transition to the endocycle and programmed cell death- but these lack statistical support). Taken together, this suggests that this protein plays some undefined role in DDR- certainly it isn't required for response, but might be required for full response. The presence of significantly higher levels of g-H2AX in both plants treated or untreated with damaging agents suggests, to me, that this protein might also be involved in gH2AX removal, though the

persistence could be due to defective repair.

There is a truly impressive amount of work in this paper, and the discovery of a new component in plant DNA repair is important. However, I had a number of issues and questions, some trivial, some important, which I'll list below. In most cases, I'm not convinced by single photos there are significant differences in for example, root length. I'd like to see better evidence that XIP and gH2AX interact. I think the quantification of spots in the HR assay need to be corrected for the amount of tissue produced (spots per gram fresh weight perhaps? Not spots per plant). I'm not sure about describing this protein as a "reader"- consider other possible roles for the putative gH2AX binding protein before deciding that it is a functional homolog of MDC1.

In Figure 3a and 3b (RT-PCR) all comparisons are to WT-Mock. But in Figure 3d (etc) does "Mock" always refer to WT's-Mock-treated? Or to the same genotype's mock-treated? The current legend: four gene sets, WT-BLM/Mock, xip-BLM/Mock, Mock-xip/WT and BLM-xip/WT is confusing to me. Same for the HU samples. Please decode and clarify.

I'm going to assume this mutant is fertile? In spite of H2AX's role in meiosis? But please tell us.

It is surprising that this mutant is not compared to (the meager) published data on the phenotype of H2AX KOs in Arabidopsis.

Data on actual protein to protein binding (as opposed to protein to peptide binding) is unconvincing and I think is limited to Fig. 1e- where gH2AX is hard to see, and doesn't seem to be in the same location as XIP. Can you demonstrate colocalization in vivo in some more convincing way?

Here are some more detailed requests:- some are just a little rewriting- others are serious issues that need to be corrected before resubmission.

L70 Is there a plant-specific reference, or is this known only in mammals and yeast. If the former, please add the ref(s). If the later, mention that this has not been directly investigated in plants. This might be a good place to discuss the phenotype of Arabidopsis H2AX mutants.

L97 This statement is highly speculative, even in the cited reference, which is simply a comparison of the radiation sensitivity of two difference species- not a generalization, and not a study of sedentary vs mobile organisms in general. I would remove this sentence, unless you want to present this idea as a speculation- here it is presented as a fact.

L116- hopefully at some point you will demonstrate that this protein actually interacts with gH2AX. Figure 1e doesn't convince me.

L124 "complemented"- maybe the authors mean "determined"?

L133 Please explain here what a "significant" vs "insignificant" BRCT domain is. (I realize this is explained in the legend to fig S1, but many readers won't get that far).

L137 "well-studied" is a very vague and subjective term. I don't know what the authors mean here. Is the dual BRCT domain found in other, poorly studied proteins? Please clarify.

L139 As in the above comment, I'm not sure why this particular set of Arabidopsis proteins was chosen for comparison. Is this a complete list of BRCT proteins in Arabidopsis?

L141- By "merely" I think you mean significant homology was observed "only" within the BRCT domain, and not elsewhere within the protein.

L151 define "ppd verification" here, don't wait until line 506

L175 Clearly the YFP has moved, but I need DAPI stain to identify the nucleus.

L177 in contrast, not "instead". Go through every use of "instead" in this paper- and fix them.

L181 The immunolocalized H2AX is very hard to see and doesn't seem to make the same shape as the YFP-tagged XIP. This is not good evidence that the two proteins colocalize in vivo.

L189 given the "slight" effects, please add some statistics, rather than just showing one photo. I wouldn't use the term "significant" without a measurement of significance (for example, root growth under normal conditions in Figure 2c doesn't seem to be significantly different from wt).

L195 it is not clear to me how the XIP gene is defined. Are you talking about XIP-M or the much longer XIP-L?

L209 Fig. 3a "X" is not called a "fork". Just call it "X".

L247 This single lane per sample western blot is not quantitative and conclusions about relative levels of expression of gH2AX aren't valid here. If this is important, perform a quantitative assay- running multiple lanes. Supplemental Figure 11 may be more convincing here.

L326 again, a single photo means nothing. Please PI staining a large number of root tips and tell us if xip has a significant effect. It's not difficult.

L342 Not sure what "in the epistasis of" means, but clearly XIP is not required for PCD. And it is not completely required for induction of the transcriptional response, though the amplitude of the response might be lessened in xip. ATM and/or ATR can still recognize DSBs without XIP.

L374 The number of spots/plant depends on the size of the plants. You're seeing about half as many spots- is this because the plants half as big? Maybe you need to normalize spots observed per gram shoot fresh weight.

Reviewer #3 (Remarks to the Author):

This is a review of the manuscript "Arabidopsis γ -H2A.X-INTERACTING PROTEIN participates in DNA damage response and safeguards chromatin stability" submitted by Fan et al. to Nature Plants. In this work, the authors aimed to identify possible novel interactors of the early DNA damage marker phosphorylated variant of histone H2A.X (γ -H2A.X). By pull-down and mass spectrometry using peptides corresponding to the non-phosphorylated and phosphorylated C-terminal region of H2A.X, they identified previously uncharacterized protein containing a pair of C-terminally located BRCT domains which they named XIP. Subsequently, the authors focused on the functional characterization of this protein. They showed that the mutants are hypersensitive to hydroxyurea and bleomycin, have problems in root apical meristem morphology upon DNA damaging treatment, and fail to fully transcriptionally activate specific DNA repair genes. Furthermore, they showed that XIP is needed for normal levels of homologous recombination. By pull-down using XIP followed by mass spectrometry, the authors identified candidates interacting with XIP, including a well-known somatic recombinase RAD51. This is an interesting story that helps understand the repair system of Arabidopsis and suggests a function for so far uncharacterized BRCT domain-containing gene. The manuscript addresses a timely topic in plant genome stability research and is generally easy to follow.

Specific comments

The plant (animal) gene and protein writing style is not followed at a number of positions in the manuscript. For example, lines 56 and 57, 72, 116

In the introduction and discussion sections, the authors mention mammalian MDI1 protein and propose XIP as its functional homolog. Although drawing analogies between proteins is tempting and may help anchor unknown proteins in the pathways, it is also tricky and can be misleading. The homology between XIP and MDI1 is over the BRCT domains. Out of curiosity, I tried BLAST with XIP protein sequence against human proteome and this revealed other proteins PTIP and PAXIP as more similar matches. That does not suggest PTIP and PAXIP as better analogs but rather indicates that making a claim of XIP being a functional homolog of MDI1 is risky and I

generally do not recommend this.

Line 137: Statement "The dual-BRCT-like domain in XIP cannot be found in any other well-studied Arabidopsis proteins" and the associated paragraph is somewhat misleading. There are certainly other Arabidopsis proteins with dual BRCT domains. I do not understand this exclusion of the known ones and the follow-up discussion related to MDI1 suffers from the same limitations as defined in my previous comment. I suggest rewriting this part to make it clearer.

Fig. 1e – the XIP signal after BLM treatment has an unusual shape. Also, the gamma-H2A.X signal is very weak and appears rather around the XIP signal than overlapping with it. Could you please clarify or provide images that provide a clearer situation?

Fig. 3b – the authors show the transcript amount of specific genes based on RT-qPCR. I suggest replacing this with data from their RNA-seq data that are normalized in a more robust way than regular RT-qPCR and moving the current Fig 3b into a supplement.

Materials and methods – I did not find anywhere a description of the used comet assay method.

Overall, based on the presented data, I would like to know where the authors place XIP in the DNA damage induction and homologous recombination pathway. Could this be presented as a model?

Line 95 – Please consider replacing "immobilized lifestyle" with "sedentary lifestyle".

Response to *Reviewers*

Reviewer #1 (Comments for the Author):

Comments:

The editor asked me to look at the quality and reliability phospho-proteomics experiments. Therefore, i have only taken a closer look at the phospho-proteomics data here.

At first I would like to give the authors here some well-meant advice. In my opinion, it would be a great idea to present the results clearly and not to overwhelm the reviewers with excel tables where it is not clear what has been done. Up to now, the technical presentation of the data has been inadequate or, for me, incomprehensible. However, a reviewer should be able to understand all the processes in the calculation of the data.

Response: We thank *Reviewer 1* for the suggestions.

Major item:

Phospho-proteomics: all relevant phospho-peptides and the corresponding non-phosphorylated peptides should be presented in a separate excel table.

Response: We thank the reviewer for the suggestion. Now in the revised manuscript, we present all the detected phosphor-peptides and proteins (for normalization) by label-free quantification phosphor-proteomic analysis in two separate excel tables as **Supplementary Table 3 and 4.**

Precursor information from Protein Discoverer search result is not very good quantitative info. It is common today that peak areas are calculated. Please calculate these peak areas with XCalibur or a quantitative software which is integrated in Protein Discoverer (e.g. qp-quant). The peak areas should be copied and displayed in a supplemental figure.

Response: We thank the reviewer for the suggestion of quantitative peak area calculation. At the same time, the reviewer also suggested a data re-analysis with *targeted proteomics (PRM)* in the final **conclusion** given below. Since parallel reaction monitoring (PRM) analysis provides more quantitative and accurate information than peak areas analysis, we chose to perform a re-analysis by PRM.

Since the **previous Supplementary Fig.11** (Selected DPPs in the four protein sets) was based on the original precursor information from Protein Discoverer search, which was considered not very good by the reviewer, we have removed this supplementary figure, and replaced it with a PRM verification as the **Supplementary Fig.12 and Table 6** in the revised manuscript. Meanwhile, all the PRM data (18.01 G), together with the previous label-free quantification phospho-proteomics data (26.76+11.2 G), have been deposited onto public database ProteomeXchange (**accession: PXD037072**).

The calculation of the phospho-peptide to non-phospho-peptide ratio should be comprehensible and the standard deviation of the peak areas should be calculated.

Response: We thank the reviewer for the suggestion. In the revised **Supplementary Table S5**, in the four samples (including WT-Mock, WT-BLM, *xip*-Mock and *xip*-BLM), the abundance of phospho-peptide was normalized to the abundance of corresponding protein (step1, shown in **red**). The average of these normalized values from three biological repeats was shown in **purple** (step 2). The comparison of these averages (such

as the ratio of WT-BLM-Average vs WT-Mock-Average, step 3, shown as *Ratio* in purple) was performed to identify the differentially phosphorylated proteins (DPPs). Those proteins harboring serine/threonine residue with up- or downregulated phospho-modification in **both fold change ≥ 2 and *p*-value < 0.05** were considered as DPPs. Since *p*-value (< 0.05) is taken into consideration in the revised manuscript but not in the previous submission, the number of DPPs is significantly decreased, and the GO analysis of these DPPs has also been updated accordingly in the revised manuscript (Fig. 4b and Supplementary Table 5).

We did our best to make the re-formatted calculation (Supplementary Table 5) clear and comprehensible. We hope all of our efforts can satisfy *Reviewer 1*.

Please note that the sequence coverage of some proteins of interest is low ($< 15\%$). Therefore, it's nearly impossible to find all potential modifications at the proteins of interest (γ -H2A.X, H2A.W.7,...).

Response: Phosphorylation is a quite dynamic post-translational modification with short half-life period. The detection of all the potential modification (especially phosphorylation here) at the examined proteins will be inevitably limited by the preparation of plant materials and the protocol of phosphor-peptide enrichment, as well as the technique and instruments used in this study. We are more concerned with the differentially phosphorylated proteins (DPPs) in *xip* when compared to WT with or without BLM treatment. We performed the enrichment and the detection of phosphorylated peptides from all the samples in parallel. Although some modification information might have been missed in this study, the findings in (phosphor-) proteomic analysis are consistent with the other findings (such as experimental histone Western) and ultimately our conclusion in this study.

Anhywere, to get a better picture of quantitative data interpretation, I opened 3 data files with Xcalibur (blm-wt, blm-xip, mock) and extracted the mass of a phosphopeptide (602.2675). Based on extracted ion chromatograms, I can see a slight upregulation (factor 2) for blm-wt.

Therefore I cannot confirm the statement of the authors (“S3 and S131 phosphorylation in H2A.W.7, which mediated DNA damage response in heterochromatin21, were also found significantly increased in xip mutant compared to WT after BLM treatment”).

Conclusion: From my side, I cannot accept this publication as it is.

To obtain more accurate or better quantification information, a re-analysis of the data with targeted proteomics (PRM) is necessary, absolutely.

Response: According to the suggestion, we have performed a data re-analysis with targeted proteomics by parallel reaction monitoring (PRM), which has greatly improved the accuracy of interested phosphor-peptide/protein abundance. All the raw-data and searching files in label-free quantification (LFQ, original data in the previous submission) and PRM analysis have been deposited onto public database ProteomeXchange (accession: PXD037072). We also compared the selected phosphorylation modification identified by LFQ and PRM (Supplementary Fig. 12), which further support the conclusion from histone Western that γ -H2A.X level has been significantly increased in *xip*-BLM when compared to WT-BLM (Fig. 4a). We hope all of our efforts in improving (phosphor-)proteomics could satisfy *Reviewer 1* during the revision stage.

The updated supplementary data:

Supplementary Table 3 and 4: All the phospho-peptides and proteins identified in the phospho-proteomic analysis in this study.

Supplementary Table 5: (re-formatted table) Normalization of phospho-peptides to the corresponding proteins, and the differentially phosphorylated proteins (DPPs) identified in this study.

Supplementary Table 6: PRM verification of phosphorylation site of selected target proteins in this study.

Supplementary Fig. 12: The comparison of phosphorylation site of selected target proteins identified by LFQ and PRM.

Reviewer #2 (Comments for the Author):

In eukaryotes, induction of a DNA DSB promotes the phosphorylation of H2AX (by localized, activated ATM) near the break to produce a zone of gamma-H2AX (γ -H2A.X). This phosphorylation is thought to act as a positive feedback loop, further activating ATM and so enhancing the DNA Damage Response, until the break is repaired and the phosphorylated H2AX removed. Plants have been shown to gamma-phosphorylate H2AX, though the significance of this event has, surprisingly, not been investigated. This paper focuses on the discovery and characterization of mutants defective in a protein that the authors describe as a “reader” for γ -H2A.X. The protein was identified as binding to a peptide matching the phosphorylated C-terminus of H2AX, but not to the peptide’s unphosphorylated version.

The term “reader” implies, to me, that it is an adaptor between γ -H2A.X and the DNA damage response, and therefore required for γ -H2A.X’s role in DDR. To determine the role of XIP in response, the authors characterize a mutant defective in this gene (named XIP). They find that some aspects of the DDR are expressed constitutively (at a mild level, in plants not intentionally exposed to exogenous DNA damaging agents), transcriptional response to DSB inducing agents still present, though perhaps muted, and some aspects of DDR may be mildly enhanced (such as premature transition to the endocycle and programmed cell death- but these lack statistical support). Taken together, this suggests that this protein plays some undefined role in DDR- certainly it isn’t required for response, but might be required for full response. The presence of significantly higher levels of γ -H2A.X in both plants treated or untreated with damaging agents suggests, to me, that this protein might also be involved in γ -H2A.X removal, though the persistence could be due to defective repair.

There is a truly impressive amount of work in this paper, and the discovery of a new component in plant DNA repair is important. However, I had a number of issues and questions, some trivial, some important, which I'll list below. In most cases, I'm not convinced by single photos there are significant differences in for example, root length. I'd like to see better evidence that XIP and γ -H2A.X interact. I think the quantification of spots in the HR assay need to be corrected for the amount of tissue produced (spots per gram fresh weight perhaps? Not spots per plant). I'm not sure about describing this protein as a "reader"- consider other possible roles for the putative γ -H2A.X binding protein before deciding that it is a functional homolog of MDC1.

Response: We thank *Reviewer 2* for the appreciation and suggestions.

In Figure 3a and 3b (RT-PCR) all comparisons are to WT-Mock. But in Figure 3d (etc) does "Mock" always refer to WT's-Mock-treated? Or to the same genotype's mock-treated? The current legend: four gene sets, WT-BLM/Mock, xip-BLM/Mock, Mock-xip/WT and BLM-xip/WT is confusing to me. Same for the HU samples. Please decode and clarify.

Response: We are sorry for the ambiguous statement in the previous manuscript. We were concerned with the significant transcriptional changes resulted only from replication or genotoxin stress in each ecotype (originally marked as HU/Mock or BLM/Mock); as well as those XIP-dependent transcriptional changes under the same condition (mock or genotoxin, originally marked as *xip*/WT). We have clarified the statement in the revised manuscript (**Fig.3c, 3d and Fig. 4b**).

WT-HU/Mock → WT-HU/WT-Mock

WT-BLM/Mock → WT-BLM/WT-Mock

xip-HU/Mock → *xip*-HU/*xip*-Mock

xip-BLM/Mock → *xip*-BLM/*xip*-Mock

Mock-*xip*/WT → *xip*-Mock/WT-Mock

HU-*xip*/WT → *xip*-HU/WT-HU

BLM-*xip*/WT → *xip*-BLM/WT-BLM

I'm going to assume this mutant is fertile? In spite of H2AX's role in meiosis? But please tell us.

Response: The *xip* single mutant has no obvious phenotype in reproduction stage. We assume that the role of *XIP* gene may be redundant with other genes and thus seemingly dispensable during meiosis. We have identified one uncharacterized XIPIP (XIP-interacting protein) with similar dual-BRCT-like domain by pulldown-MS analysis (Supplementary Table 7). Our unpublished genetic analysis has revealed that this double mutant has a slight but significant defect in fertility. We hope to unravel the coordination of XIP with other DDR protein in the meiotic homologous recombination and plant meiosis in the future analysis.

[Collected]

Data on actual protein to protein binding (as opposed to protein to peptide binding) is unconvincing and I think is limited to Fig. 1e- where γ -H2A.X is hard to see, and doesn't seem to be in the same location as XIP. Can you demonstrate colocalization in vivo in some more convincing way?

L116- hopefully at some point you will demonstrate that this protein actually interacts with γ -H2A.X. Figure 1e doesn't convince me.

L181 The immunolocalized H2AX is very hard to see and doesn't seem to make the same shape as the YFP-tagged XIP. This is not good evidence that the two proteins colocalize in vivo.

Response: We are sorry that the previous image was not solid enough to support the conclusion that XIP acts as an H2A.X-interacting protein *in vivo*.

We have optimized our co-immunostaining experiment, and here in the revised manuscript, we provided a convincing result that XIP signals are co-localized with BLM-induced γ -H2A.X foci in the DAPI-stained nuclei (**Fig. 1e**).

Here are some more detailed requests:- some are just a little rewriting- others are serious issues that need to be corrected before resubmission.

[Collected]

L70 Is there a plant-specific reference, or is this known only in mammals and yeast. If the former, please add the ref(s). If the later, mention that this has not been directly investigated in plants. This might be a good place to discuss the phenotype of Arabidopsis H2AX mutants.

It is surprising that this mutant is not compared to (the meager) published data on the phenotype of H2AX KOs in Arabidopsis.

Response: Arabidopsis genome has two redundant genes encoding H2A.X. Although dispensable for plant viability and fertility, the mutants and silenced lines of *H2A.X* have been reported as mildly hypersensitive to genotoxin and defective in DSB repair (Huefner et al., 2009; Lang et al., 2012; Qi et al., 2016; Waterworth et al., 2019). We thank for the suggestions. In the revised manuscript, we have clarified the citation (*L70*) and added the introduction to the Arabidopsis *h2a.x* mutants (reference 10~13), which

facilitates the functional comparison of *XIP* with *H2A.X*.

Supplementary Reference:

Huefner ND, Friesner JD, Britt AB. Characterization of Two H2AX Homologues in *Arabidopsis thaliana* and their Response to Ionizing Radiation. *Induced Plant Mutations in the Genomics Era*, 113-117 (2009).

Lang J, et al. Plant gammaH2AX foci are required for proper DNA DSB repair responses and colocalize with E2F factors. *New Phytol* **194**, 353-363 (2012).

Qi Y, Zhang Y, Baller JA, Voytas DF. Histone H2AX and the small RNA pathway modulate both non-homologous end-joining and homologous recombination in plants. *Mutat Res* **783**, 9-14 (2015).

Waterworth WM, et al. Phosphoproteomic analysis reveals plant DNA damage signalling pathways with a functional role for histone H2AX phosphorylation in plant growth under genotoxic stress. *Plant J* **100**, 1007-1021 (2019).

L97 This statement is highly speculative, even in the cited reference, which is simply a comparison of the radiation sensitivity of two difference species- not a generalization, and not a study of sedentary vs mobile organisms in general. I would remove this sentence, unless you want to present this idea as a speculation- here it is presented as a fact.

Response: We thank for the suggestion and have removed this reference in the revised manuscript.

L124 “complemented”- maybe the authors mean “determined”?

Response: We thank for the suggestion and have changed the word in the revised manuscript.

L133 Please explain here what a "significant" vs "insignificant" BRCT domain is. (I realize this is explained in the legend to fig S1, but many readers won't get that far).

Response: We thank for the suggestion and have added the necessary explanation of “(in)significance” in the revised manuscript.

L137 "well-studied" is a very vague and subjective term. I don't know what the authors mean here. Is the dual BRCT domain found in other, poorly studied proteins? Please clarify.

Response: In our knowledge, the certain dual-BRCT-like domain (a significant RTT107_BRCT_5 domain/PF16770 followed by an insignificant BRCT domain/PF00533) have not been found in any characterized **Arabidopsis proteins with reported functions**. However, the dual-BRCT-like domain can be also found in one XIPIP identified in this study (encoded by AT2G41450), which *has not been studied yet*. We plan to perform further molecular and functional characterization of these dual-BRCT-like domain-containing proteins in the future. We have removed the ambiguous statement in the revised manuscript.

L139 As in the above comment, I'm not sure why this particular set of Arabidopsis proteins was chosen for comparison. Is this a complete list of BRCT proteins in Arabidopsis?

Response: As previously reported (Singh et al., 2008), 34 BRCT-like domains have

been detected in 28 different proteins in Arabidopsis. However, nearly half of these proteins have not been characterized and remained functionally unknown. As a result, we selected all the **BRCT-like domain-containing proteins with reported functions**, including well-known heterodimer AtBRCA1-AtROW1 and DNA repair proteins AtLIG4 and AtPARP1 (previous Supplementary Fig.2) to construct the phylogenetic tree. Our previous intention in the initial submission is to highlight the certain dual-BRCT-like domain (a significant RTT107_BRCT_5 domain/PF16770 followed by an insignificant BRCT domain/PF00533) and its novel function, which cannot be found in the proteins previously reported.

Due to the limitation of functional characterization of all the BRCT-like domain-containing proteins, the current phylogenetic tree cannot be comprehensive, and prone to result in misleading. Now, in the revised manuscript, we removed the phylogenetic tree (previous Supplementary Fig.2, now removed), which seems unnecessary and dispensable in the interpretation of XIP function in this study.

Supplementary Reference:

Singh SK, Choudhury SR, Roy S, Sengupta DN. Sequential, structural, and phylogenetic study of BRCT module in plants. *Journal of biomolecular structure & dynamics* **26**, 235-245 (2008).

L141- By “merely” I think you mean significant homology was observed “only” within the BRCT domain, and not elsewhere within the protein.

Response: We thank for the suggestion.

L151 define “ppd verification” here, don’t wait until line 506

Response: In the original manuscript, we abbreviated the “peptide pulldown” as **ppd** first in *L121*, and the ppd experiment has been introduced in *L110-L113*. Briefly, peptides (p1/p2/p3) were N-terminally biotinylated so as to be immobilized by Streptavidin-coated magnetic beads, and were used to retrieve interacting proteins from Arabidopsis protein extracts for *mass spectrum* (Fig. 1a and 1b) and from recombinant proteins for *verification* (Supplementary Fig. 3).

L175 Clearly the YFP has moved, but I need DAPI stain to identify the nucleus.

Response: In the revised manuscript, we have provided the co-localization of YFP-XIP with γ -H2A.X foci in the immunostaining assay, in which DAPI staining was used to label the nucleus (Fig. 1e). Our observation has shown a clear **nuclear localization** of YFP-XIP after genotoxic stress (indicated by both γ -H2A.X and DAPI).

L177 in contrast, not “instead”. Go through every use of “instead” in this paper- and fix them.

Response: We thank for the suggestion.

L189 given the "slight" effects, please add some statistics, rather than just showing one photo. I wouldn't use the term “significant” without a measurement of significance (for example, root growth under normal conditions in Figure 2c doesn't seem to be significantly different from wt).

Response: Fig. 2c displayed the direct comparison of **WT primary root length** with

or without genotoxin treatment. The original data of root length can be found in the source data in the revised manuscript, and the significance of these inhibition has been confirmed by the *t*-test, which indicates the validity of the replicative and genotoxic stresses in this study.

We speculate that “root growth without a measurement of significance in Fig. 2c” mentioned by reviewer may be the one in **Fig. 2b**. Now, we quantified the meristem size by measuring the length of proximal meristem (in a cortex cell file extending from the quiescent center to the first elongated cortex cell) according to a previous study (Perilli and Sabtini, 2010). The quantified data have been added in the supplementary data as **Supplementary Fig. 7** in the revised manuscript, and the source data have also been provided.

In addition, as mentioned below, *reviewer 2* has suggested to normalize HR spots observed per gram shoot fresh weight. As a result, we performed new HRF measurement, and weighted the seedlings at the same time. Actually, the growth of *xip* (indicated by the wet weight) was inhibited both under normal growth condition and after genotoxin stress. We have provided these data in the source data file.

Supplementary Reference:

Perilli S, Sabatini S. Analysis of root meristem size development. *Methods Mol Biol* **655**, 177-187 (2010).

L195 it is not clear to me how the XIP gene is defined. Are you talking about XIP-M or the much longer XIP-L?

Response: In *L195*, the entire genomic DNA of *XIP* includes 2 kb of promoter and 1 kb of terminator of *XIP-L*. We have clarified this information in the revised manuscript.

L209 Fig. 3a “X” is not called a “fork”. Just call it “X”.

Response: We thank for the suggestion.

L247 This single lane per sample western blot is not quantitative and conclusions about relative levels of expression of γ -H2A.X aren't valid here. If this is important, perform a quantitative assay- running multiple lanes. Supplemental Figure 11 may be more convincing here.

Response: We found that the relative level of γ -H2A.X (referenced to H3) was obviously increased in *xip*-BLM when compared to WT-BLM in Western blot (Fig. 4a). In the revised manuscript, we have provided the quantitative data of relative levels of γ -H2A.X (referenced to H3.1, the canonical histones of H3) by LFQ and PRM (Supplementary Fig. 12 and Table 6). Both quantitative phosphor-proteomic analyses further support our conclusion that γ -H2A.X level is increased in *xip* background when compared to WT.

L326 again, a single photo means nothing. Please PI staining a large number of root tips and tell us if xip has a significant effect. It's not difficult.

Response: 15-20 root tips per sample have been observed in our PI staining assay, and the typical images of stained root tips have been shown. Now, the number of observed root tips was added in Fig.5a and Supplementary Fig.13b in the revised manuscript to highlight the significant effects.

L342 Not sure what "in the epistasis of" means, but clearly XIP is not required for PCD. And it is not completely required for induction of the transcriptional response, though the amplitude of the response might be lessened in xip. ATM and/or ATR can still recognize DSBs without XIP.

Response: In this study, more genome instability, more DNA fragment, and more cell death were found in *xip* mutant when compared to WT. The PI staining result in **Fig.5a** clearly indicates that increased cell death caused by XIP depletion is dependent on SOG1 activity. At the same time, the double mutant *sog1-101 xip* [Col] showed similar transcriptional response with *sog1-101* (**Fig.5b**).

Since more genetic and molecular crosstalk may be needed to delineate the pathway including both XIP and SOG1, we have adjusted the text in the revised manuscript to avoid any potential overstatement.

L374 The number of spots/plant depends on the size of the plants. You're seeing about half as many spots- is this because the plants half as big? Maybe you need to normalize spots observed per gram shoot fresh weight.

Response: We thank for the suggestion. Now we have performed another HRF assay, and counted the HRF and measured the weight of examined seedlings. All the spots/sections per plant were further normalized to fresh weight (mg). Our data indicated that HRFs revealed by two HR markers, *1445* and *IC9C*, were both significantly down-regulated in *xip* when compared to WT, indicative of the important contribution of XIP to somatic HR occurrence. By the way, the data of the fresh weight also facilitate the quantitative comparison of *xip* seedling growth with WT (for **Fig.2a**)

Reviewer #3 (Comments for the Author):

This is a review of the manuscript “Arabidopsis γ -H2A.X-INTERACTING PROTEIN participates in DNA damage response and safeguards chromatin stability” submitted by Fan et al. to Nature Plants. In this work, the authors aimed to identify possible novel interactors of the early DNA damage marker phosphorylated variant of histone H2A.X (γ -H2A.X). By pull-down and mass spectrometry using peptides corresponding to the non-phosphorylated and phosphorylated C-terminal region of H2A.X, they identified previously uncharacterized protein containing a pair of C-terminally located BRCT domains which they named XIP. Subsequently, the authors focused on the functional characterization of this protein. They showed that the mutants are hypersensitive to hydroxyurea and bleomycin, have problems in root apical meristem morphology upon DNA damaging treatment, and fail to fully transcriptionally activate specific DNA repair genes. Furthermore, they showed that XIP is needed for normal levels of homologous recombination. By pull-down using XIP followed by mass spectrometry, the authors identified candidates interacting with XIP, including a well-known somatic recombinase RAD51. This is an interesting story that helps understand the repair system of Arabidopsis and suggests a function for so far uncharacterized BRCT domain-containing gene. The manuscript addresses a timely topic in plant genome stability research and is generally easy to follow.

Response: We thank *Reviewer 3* for the encouragement.

Specific comments

The plant (animal) gene and protein writing style is not followed at a number of positions in the manuscript. For example, lines 56 and 57, 72, 116

Response: We introduced the name of gene/protein by the acronym following the full

name in the manuscript. We are sorry that it may be a little confusing when the full name is comprised of another acronym. The full name of ATR is ATM and Rad3-related, in which ATM is the acronym of ataxia telangiectasia mutated. The full name of BRCT is BRCA1 carboxyl-terminal, in which BRCA1 is the acronym of breast cancer associated 1. We have checked the whole manuscript and confirmed the consistence of writing style of mentioned gene/protein names.

In the introduction and discussion sections, the authors mention mammalian MDII protein and propose XIP as its functional homolog. Although drawing analogies between proteins is tempting and may help anchor unknown proteins in the pathways, it is also tricky and can be misleading. The homology between XIP and MDII is over the BRCT domains. Out of curiosity, I tried BLAST with XIP protein sequence against human proteome and this revealed other proteins PTIP and PAXIP as more similar matches. That does not suggest PTIP and PAXIP as better analogs but rather indicates that making a claim of XIP being a functional homolog of MDII is risky and I generally do not recommend this.

Response: Since XIP is the first protein identified to interact specifically with γ -H2A.X but not the unphosphorylated H2A.X, and XIP share certain homology with MDC1 only in dual-BRCT-like domains, we proposed that XIP may act as a functional **analog** but not homolog of MDC1 in the original manuscript. Here, in the revised manuscript, we have removed the relevant statement to weaken the direct comparison of XIP with MDC1.

Line 137: Statement “The dual-BRCT-like domain in XIP cannot be found in any other well-studied Arabidopsis proteins” and the associated paragraph is somewhat misleading. There are certainly other Arabidopsis proteins with dual BRCT domains. I

do not understand this exclusion of the known ones and the follow-up discussion related to MDII suffers from the same limitations as defined in my previous comment. I suggest rewriting this part to make it clearer.

Response: As mentioned above, in our knowledge, the certain dual-BRCT-like domain (a significant RTT107_BRCT_5 domain/PF16770 followed by an insignificant BRCT domain/PF00533) have not been found in any characterized Arabidopsis proteins with **reported functions**. However, the dual-BRCT-like domain can be found in one XIPIP identified in this study (encoded by AT2G41450), which *has not been studied yet*. We plan to perform further molecular and functional characterization of these dual-BRCT-like domain-containing proteins in the future.

In the revised manuscript, we have removed the misleading phylogenetic tree (**previous Supplementary Fig.2, now removed**) and the corresponding statement. We keep the conclusion that XIP has low similarity in protein sequence, except dual-BRCT-domain, with animal MDC1, and we discussed the common and distinct aspects of these two γ -H2A.X-interacting proteins in DDR pathways.

Fig. 1e – the XIP signal after BLM treatment has an unusual shape. Also, the gamma-H2A.X signal is very weak and appears rather around the XIP signal than overlapping with it. Could you please clarify or provide images that provide a clearer situation?

Response: We are sorry that the previous image is not solid enough to support the conclusion that XIP acts as an H2A.X-interacting protein *in vivo*.

We have optimized our co-immunostaining experiment, and here in the revised manuscript, we provided a convincing result that XIP signals are co-localized with BLM-induced γ -H2A.X foci in DAPI-stained nuclei.

Fig. 3b – the authors show the transcript amount of specific genes based on RT-qPCR. I suggest replacing this with data from their RNA-seq data that are normalized in a more robust way than regular RT-qPCR and moving the current Fig 3b into a supplement.

Response: We have extracted the normalized FPKM (Fragment Per Kilobase of transcript per Million fragments mapped) of all the examined genes shown in **Fig. 3a/3b** from RNA-seq, which showed a similar result with our RT-qPCR.

The **Fig. 3c/3d** is composed of analyzed data based on the RNA-seq. Since RT-qPCR is an experimental verification to the high-throughput sequencing, we hope to remain the RT-qPCR results in the original **Fig. 3a/3b**, and the extracted FPKM are placed in the **Supplementary Fig. 9** in the revised manuscript.

Materials and methods – I did not find anywhere a description of the used comet assay method.

Response: We now provide a description of comet assay in the revised manuscript.

Overall, based on the presented data, I would like to know where the authors place XIP in the DNA damage induction and homologous recombination pathway. Could this be presented as a model?

Response: In this study, we identified XIP as a protein specifically interacting with γ -H2A.X, and found that it can also interact with RAD51, the key recombinase involved in homologous recombination (HR) pathway, which facilitates the HR occurrence for DSB repair. Our genetic and phenotypic analyses revealed that XIP depletion causes transcriptional mis-regulation and abnormal cell death, which require SOG1 activity to

mediate the DDR pathway. However, till now, it is not easy to place XIP accurately in the complicate network of plant DNA damage sensing, signaling and repair pathways. Many uncharacterized plant-specific players may participate in these pathways, which cannot be explained by simply invoking the mechanisms unraveled in yeast and animal cells. More investigations in the future are needed to provide enough clues to establish a plant-specific DDR network.

Line 95 – Please consider replacing “immobilized lifestyle” with “sedentary lifestyle”.

Response: We thank for the suggestion.

Reviewer #1 (Remarks to the Author):

Reviewer 1:

I would like to thank the authors very much that the phospho-proteomics data are now presented in a much better and transparent way.

If the authors have time, they can re-analyze the data again with more PRM transitions. This would improve the precision and accuracy of the quantification. However, it is not necessary to do this work, as supplementary figure 12 is very informative.

In my research group, we also deal with the same issues and verify important phospho-sites with a second independent method, which helps to significantly improve the statistics of the analysis (power of analytics).

I have looked at the data very well and if the authors have time they can include a statement in supplementary information.

Question 1:

In the shotgun analysis, the phosphorylated peptide is found 2 times for the protein AT1G08880.1. Once with and once without miss-cleavage (supplementary table 3 line 307/308).

Sequence : NK-GDIGSASQEF and GDIGSASQEF

The peptide with the miss-cleavage (NK-GDIGSASQEF) shows a much higher peak area in the shotgun analysis in comparison to the normal peptide (ratio of 6 at the sample xip-BLM-2).

As a consequence of the small peak area, the peptide was not found without miss-cleavage in 50% of all experiments.

If I were to commission the experiment to my staff, I would request all 2 transitions in the analysis. I therefore do not understand why the authors only used the peptide with the smaller peak area for PRM analysis.

I therefore wonder why all 2 peptides were not used in the PRM analysis. Or were all 2 peptides used in the prm analysis and not evaluated? I ask because I did not have time to look at the RAW data.

Question 2:

The protein AT1G26630.1 was also found to have the phosphorylated peptide 2 times (line 1097 and 1098). Once N-terminally acetylated and once not acetylated. Again, I would have a similar question, why not use both peptides for the PRM analysis to improve the statistical conclusion.

If I now calculate the ratio of the acetylated peptide to the non-acetylated peptide, I arrive at a ratio of 22:1 to 53:1.

1279627 2462737 1327350 4269131 813720 1706684 1402693 4377390 2457532 1269637
1554042 3407245

48020200 54045421 51257094 94889924 43421545 44033091 43387321 81704286 39672547
39691319 33901606 63518403

37.5 21.9 38.6 22.2 53.4 25.8 30.9 18.7 16.1 31.3 21.8 18.6

Question 3:

Similar question to protein AT4G35230.1: the peptide was detected 3 times (non, miss-cleavage, pyro-glu).

Question 4:

For the normalization of the protein ratio, we work with a minimum of 3 peptides, where we always mixing the sequences of the non-phospho-peptide as well.

For the normalization of the total proteome, we use 3 background proteins, taking care that these proteins cannot be excessively modified. Therefore, histones are not so well suited.

Summary:

I would like to thank the authors for supplementary figure 12, which shows the comparison of the selected phosphorylation modification identified by LFQ and PRM. Supplementary figure 12 support

the conclusion from histone Western that γ -H2A.X level has been significantly increased in xip-BLM when compared to WT-BLM.

I would like to thank you very much for the re-analysis of the data. If the ptm-rs module was used in proteome discoverer 2.4 workflow, it should be cited accordingly.

Reviewer #2 (Remarks to the Author):

This is an extensively revised version of a previous paper.

The authors performed a search for Arabidopsis proteins that specifically interact with a peptide based on the phosphorylated version of H2AX (and not the non-phosphorylated peptide). They come up with a single gene that has not been previously described and proceed to investigate some activities of the protein (its localization, its interactions with other proteins) and many aspects of the phenotype of a KO mutant, including effects on the transcriptome and phosphatome. There's a lot of data here, which is great, but there still are many issues with interpretation, and lesser issues with description of the methods and presentation of the data. The phenotype is subtle and it is difficult to draw any conclusions as to what this protein actually does. Conclusions are often underthought and overstated (and therefore misleading to many readers). The authors, in the previous version, felt it was a MDC1 homolog, and they may have changed their minds in this version, but there's still a lot of vestigial discussion of MDC1.

In mammals, MDC1 interacts with phosphorylated gH2AX and seems to be required for the "interpretation" of this signal- the phenotype of the MDC1 KO is very similar to that of an H2AX KO. But this protein, XIP, does not resemble MDC1 (except in its dual BRCT domains, and many proteins with various repair- or response related functions, have this dual BRCT domain). In their "rebuttal" the authors state that they regard this protein as a "functional analog" of MDC1- though they don't actually state this in the paper (in fact, they continue to imply it is a homolog), nor do they provide any evidence suggesting that this protein performs the same function as MDC1 - other than its binding to gH2AX. In the KO mutant the damage signal still clearly gets through to the plant transcriptome. The authors claim that the response is slightly muted, but some of this "muting" effect on the number of genes induced or repressed by damage may be due to the constitutive, low level, induction of the DDR response in even untreated mutant plants. The authors need to keep in mind that there are many, many ways to disrupt DNA metabolism- constitutive low level DNA damage response and genome instability can result from errors in nucleotide pool regulation, redox regulation, DNA replication, transcription, repair, and, finally, damage response. One mutator mutant (mutQ) in E coli is actually a mis-loading tRNA, so add errors of translation to that list too.

What I think we have here is a new, previously undescribed gene, encoding a protein that probably binds H2AX, probably only in its phosphorylated form, and also binds RAD51. Plants carrying a KO of XIP grow a little more slowly than wt, are slightly more sensitive to the effects of DNA damaging agents on growth, and have a constitutive but low level transcriptional damage response- suggesting there is a constitutively higher than normal level of endogenous DNA damage. There's no reason to conclude that this is a response defect rather than a repair defect. I have no expertise in phosphopeptide analysis, so I'm not discussing that.

My summary is that this is no doubt a gene that is somehow involved in DNA repair or response, but although the authors have put in an exceptional effort, we really have no idea of what this gene does. As the authors state in their rebuttal, there's really no place to put this gene in a diagram of the response or repair pathways.

Here's more detailed analysis of what I still find unconvincing (in terms of the data), and what I find "solid". Plus suggestions on English, which I think is mostly understandable, though it could use some cleaning up.

L35 clarify whether you mean the response is depressed or enhanced, if you have a conclusion.

L39 The conclusion that XIP is required to relay the H2AX signal is in my opinion completely unjustified. Again, as I stated in my last review, a mutant required to relay the signal would match the phenotype of a mutant completely defective in gH2AX- the authors don't do this experiment. See also line 100.

L139 The authors again use the term "insignificant" and tell me in the rebuttal that that's what the Pfam program labeled it- they still don't define it. Please let us know what Pfam is trying to tell us with this term. It's a weird way to describe a domain that's essential for the one function that you know this protein performs- binding of your peptide.

L143 "is close to...MDC1" is a meaningless term- you've already told us your domain is a dual BRCT, and as mentioned by another reviewer, it is "closer" to the dual domains of other human proteins. I would stop trying to cram this protein into the MDC1 homolog/analog box and view it with fresh eyes for what it is- a protein of unknown function.

L149 "unexpectedly low similarity in primary protein sequence may explain why clear MDC1 homologs have not been found in previous plant studies". This outrageous statement clearly states that you have found the homolog in THIS study, which you have not. There is zero homology outside the BRCT domains, which are a frequently-employed motif.

L178 What on earth is going on with the cytoplasm/vacuole in the treated protos? Has the vacuolar membrane been lost? Is it always like this after treatment and never like this in untreated protos? If the cytoplasm and vacuole are mixed, the signal might simply be too dilute to see. So it's not clear that the protein has migrated or simply been diluted in this vacuolar/cytoplasmic mix.

L185 Fig. 1e does show one photo in which the two labels overlay- though the foci are not very obvious. Please also show us an untreated cell.

L187 I agree with everything in this paragraph! But please clarify the nature of the treatment- for example, how long were the plants exposed to the damaging agent? Provide enough detail for another lab to repeat your experiment.

L212 again, we don't know the duration of the treatment

L215 drop "much"- that's meaningless, especially as this was significant only at <0.05

L221 replace "all the examined" repair genes with "all three examined"

L236 You're comparing the number of genes induced at least 2x, with a certain P value, in treated vs untreated mutants. Your xip control/wt control data already tells you that you have low-level constitutive expression of DDR in xip untreated (which is the most convincing proof that these mutants are experiencing constitutive genomic stress). But this might also explain why your treated xip plants do not have as many new genes induced by damage as wt plants, because they are already overexpressing these genes to some degree, so their fold induction is less significant.

L282 Quantitation of gH2AX is useful but you are leaping to a single conclusion- not considering alternative hypotheses. More gH2AX may be present because:

There are more breaks, as you suggest, but also, instead...

gH2AX is more persistent at each break (it is not being unloaded after repair)

gH2AX is more extensively loaded at each break

and you can't distinguish between

more breaks generated by your DNA damaging agent (which you suggest but frankly seems unlikely? what would be the mechanism?)

breaks are persisting longer, i.e. slow repair

L312 I'm not a fan of the comet assay for anything but the most obvious differences- especially in plants, and especially when they look like comet "hedgehogs". I think the transcriptional response is more convincing as evidence of instability. They might also consider a sectoring assay for chromosome arm loss.

L322 I do like the PI assay for cell death but the authors need to tell us how they quantified this.

They again show us single photos and draw remarkable conclusions from images that look very similar (1 dead cell in a hundred is not significantly different from zero dead cells in a hundred).

The images do have a tiny notation in the corner, for example, 20/20 or 17/20. They need to tell us what these mean- I'm guessing they just scored for presence or absence of dead cells, and didn't count the frequency of dead cells in each root tip, but help me out here.

L376 again, misleading overstatement. XIP is not required for HR activity- it still occurs. It is somewhat reduced- that's different.

L377 "The finding of decreased HRF was also consistent with the increased chromatin instability".

Again, think of additional hypotheses- decreased somatic mutagenesis (here, the homologous repair of spontaneous breaks at the test locus) is ALSO consistent with DECREASED rates of spontaneous breaks. Both are true, so it's not a very helpful assay.

L383 why "unique"? You hadn't mentioned "uniqueness" before. Do you just mean "a gene no one has discussed before"? If so drop the term.

L386 again, "required for cell cycle progression" is a HUGE overstatement, both in terms of

phenotype and specificity of function. The mutants are slightly smaller than wt. Cell cycle progression is still happening. There's no evidence that this gene is at all involved in cell cycle progression as opposed to virtually any housekeeping function required for optimal growth. Rethink this sentence.

L388 again, you have no idea at all what the function of this gene is. It has an impact on DDR- that does not really mean its function is to regulate DDR. Virtually anything that impacts DNA metabolism- i.e., affecting nucleotide pools, replication, transcription, repair, or response, will impact genome stability.

L444 please don't tell us about this mystery protein until you have published data on it.

Reviewer #3 (Remarks to the Author):

I would like to thank the authors for successfully addressing most of my concerns. However, there are still several issues remaining.

The plant nomenclature of gene and protein names uses capital letters for the WT variants and small letters for mutant versions (+ italics for genes and non-italics for proteins). Therefore, „ATM and Rad3-related“ should be „ATM AND RAD3-RELATED“. I recommend fixing this throughout the manuscript.

I acknowledge the authors' response about their view on the analogy between XIP and MDC1. However, I still do not understand their absolute preference towards human MDC1 (mentioned 74 times in the manuscript) while entirely neglecting (0 mentions) PTIP and PAXIP with higher homology to BRCT domains of Arabidopsis XIP protein. Although the authors write that they weakened their statement, the current version of the manuscript still offers MDC1 as a single option (see discussion part „ γ -H2A.X recognition by XIP and MDC1“). In short, I do not deny that XIP is a functional analog of human MDC1, but I am still missing strong argumentation why this is the best option.

As mentioned in the first round of revisions, I am also missing a model or discussion about the position of XIP in the DSB repair pathway.

Response to *Reviewers*

Reviewer #1 (Comments for the Author):

I would like to thank the authors very much that the phospho-proteomics data are now presented in a much better and transparent way.

If the authors have time, they can re-analyze the data again with more PRM transitions. This would improve the precision and accuracy of the quantification. However, it is not necessary to do this work, as supplementary figure 12 is very informative.

In my research group, we also deal with the same issues and verify important phospho-sites with a second independent method, which helps to significantly improve the statistics of the analysis (power of analytics).

I have looked at the data very well and if the authors have time they can include a statement in supplementary information.

Response: We thank *Reviewer 1* for the appreciation and suggestions.

Question 1:

In the shotgun analysis, the phosphorylated peptide is found 2 times for the protein AT1G08880.1. Once with and once without miss-cleavage (supplementary table 3 line 307/308).

Sequence : NK-GDIGSASQEF and GDIGSASQEF

The peptide with the miss-cleavage (NK-GDIGSASQEF) shows a much higher peak area in the shotgun analysis in comparison to the normal peptide (ratio of 6 at the

sample xip-BLM-2).

As a consequence of the small peak area, the peptide was not found without miss-cleavage in 50% of all experiments.

If I were to commission the experiment to my staff, I would request all 2 transitions in the analysis. I therefore do not understand why the authors only used the peptide with the smaller peak area for PRM analysis.

I therefore wonder why all 2 peptides were not used in the PRM analysis. Or were all 2 peptides used in the prm analysis and not evaluated? I ask because I did not have time to look at the RAW data.

Response: We thank *Reviewer 1* for the suggestion. Actually, we found three phospho-peptides and one protein (non-phospho-)peptide of AT1G08880.1 (H2A.X) in the PRM analysis, respectively. Two phospho-peptides are H2A.X-S139, and another is H2A.X-S137, which is irrelevant to γ -H2A.X and has not been analyzed further (see below).

	positions in master proteins	Phospho-peptides	Phospho-sites	The analyzed phospho-peptides
phospho-peptide	AT1G08880.1 [133-142]	GDIGSAS[Phospho (STY)]QEF	H2A.X-S139	phospho-peptide 1
	AT1G08880.1 [131-142]	NKGDIGSAS[Phospho (STY)]QEF	H2A.X-S139	phospho-peptide 2
	AT1G08880.1 [131-142]	NKGDIGS[Phospho (STY)]ASQEF	H2A.X-S137	another phospho-site, not used here
			H2A.X-S139	phospho-peptide-sum (1+2)
	positions in master proteins	Peptide used for quantification	ProteinName	
protein	AT1G08880.1 [131-142]	NKGDIGSASQEF	H2A.X	protein H2A.X
			Normalization	
			H2A.X-S139	phospho-peptide 1/protein H2A.X
			H2A.X-S139	phospho-peptide 2/protein H2A.X
			H2A.X-S139	phospho-peptide-sum/protein H2A.X

We normalized the two H2A.X-S139 phospho-peptides and their sum (1+2), respectively, to H2A.X protein (there is only one protein peptide of H2A.X identified in this study, so there is no sum), which reached similar results. Now, we show the phospho-peptide-sum/protein H2A.X instead of the previous 'phospho-peptide 1/protein H2A.X' in Supplementary Fig.13 in the revised manuscript.

In addition, we normalized the two H2A.X-S139 phospho-peptides and their sum (1+2), respectively, to the protein sum of histone H3.1, which reached similar results. Now, we show the phospho-peptide-sum/protein H3.1-sum instead of the previous ‘phospho-peptide 1/protein H3.1-1’ in Supplementary Fig.13 in the revised manuscript.

	positions in master proteins	Phospho-peptides	Phospho-sites	The analyzed phospho-peptides
phospho-peptide	AT1G08880.1 [133-142]	GDIGSAS[Phospho (STY)]QEF	H2A.X-S139	phospho-peptide 1
	AT1G08880.1 [131-142]	NKGDIGSAS[Phospho (STY)]QEF	H2A.X-S139	phospho-peptide 2
	AT1G08880.1 [131-142]	NKGDIGS[Phospho (STY)]ASQEF	H2A.X-S137	another phospho-site, not used here
			H2A.X-S139	phospho-peptide-sum (1+2)
	positions in master proteins	Peptide used for quantification	ProteinName	
protein	AT1G09200.1 [44-50]	PGTVALR	H3.1	protein H3.1-1
	AT1G09200.1 [29-37]	SAPATGGVK	H3.1	protein H3.1-2
			H3.1-sum	protein-H3.1-sum
Normalization				
			H2A.X-S139/H3.1-sum	phospho-peptide 1/protein H3.1-sum
			H2A.X-S139/H3.1-sum	phospho-peptide 2/protein H3.1-sum
			H2A.X-S139-sum/H3.1-sum	phospho-peptide-sum/protein H3.1-sum

All the data and the detail of normalization can be found in the Supplementary Table 6 in the revised manuscript.

Question 2:

The protein AT1G26630.1 was also found to have the phosphorylated peptide 2 times (line 1097 and 1098). Once N-terminally acetylated and once not acetylated. Again, I would have a similar question, why not use both peptides for the PRM analysis to improve the statistical conclusion.

If I now calculate the ratio of the acetylated peptide to the non-acetylated peptide, I arrive at a ratio of 22:1 to 53:1.

*1279627 2462737 1327350 4269131 813720 1706684 1402693 4377390 2457532
1269637 1554042 3407245*

*48020200 54045421 51257094 94889924 43421545 44033091 43387321 81704286
39672547 39691319 33901606 63518403*

37.5 21.9 38.6 22.2 53.4 25.8 30.9 18.7 16.1 31.3 21.8 18.6

Response: Similarly, we found three phospho-peptides and three protein (non-phospho-)peptides of AT1G26630.1 (FBR12) in the PRM analysis, respectively. All three phospho-peptides are FBR12-S2, and notably, as *Reviewer 1* mentioned, the

abundance of phospho-peptides 2 and 3 (both are non-acetylated) are negligible when compared to phospho-peptide 1 (acetylated). As a result, we labelled phospho-peptide 1 as ‘major’ and phospho-peptide 2/3 as ‘minor’ (see below).

	positions in master proteins	Phospho-peptides	Phospho-sites	The analyzed phospho-peptides
phospho-peptide	AT1G26630.1 [2-17]	[Acetyl (Protein N-term)]S[Phospho (STY)]DDEH	FBR12-S2	phospho-peptide 1, major
	AT1G26630.1 [2-17]	S[Phospho (STY)]DDEHHFEASESGASK	FBR12-S2	phospho-peptide 2, minor
	AT1G26630.1 [2-17]	S[Phospho (STY)]DDEHHFEASESGASK	FBR12-S2	phospho-peptide 3, minor
			FBR12-S2-sum	phospho-peptide-sum (1+2+3)
	positions in master proteins	Peptide used for quantification	ProteinName	
protein	AT1G26630.1 [18-27]	TYPQSAGNIR	FBR12	protein 1
	AT1G26630.1 [116-127]	LPTDDGLTAQMR	FBR12	protein 2
	AT1G26630.1 [112-127]	DDLKLPDDGLTAQMR	FBR12	protein 3
			FBR12-sum	protein-sum
Normalization				
			FBR12-S2/FBR12-sum	phospho-peptide 1/protein-sum
			FBR12-S2/FBR12-sum	phospho-peptide 2/protein-sum
			FBR12-S2/FBR12-sum	phospho-peptide 3/protein sum
			FBR12-S2-sum/FBR12-sum	phospho-peptide-sum/protein-sum

Therefore, we normalized the phospho-peptide 1 and the sum (1+2+3) of three phospho-peptides to the protein sum of three peptides, respectively, which reached similar results. Now, we show the phospho-peptide-sum/protein-sum instead of the previous ‘phospho-peptide-1 (major)/protein-1’ in **Supplementary Fig.13** in the revised manuscript.

All the data and the detail of normalization can be found in the revised **Supplementary Table 6**.

Question 3:

Similar question to protein AT4G35230.1: the peptide was detected 3 times (non, miss-cleavage, pyro-glu).

Response: Similarly, we found four phospho-peptides and three protein (non-phospho-)peptides of AT4G35230.1 (BSK1) in the PRM analysis, respectively. Two phospho-peptides are BSK1-S353, and the other two are not the same sites (S228 and S230), which are irrelevant to the corresponding LFQ analysis and have not been analyzed further (see below).

	positions in master proteins	Phospho-peptides	Phospho-sites	The analyzed phospho-peptides
phospho-peptide	AT4G35230.1 [346-367]	KQEEAPST[Phospho (STY)]PQRPLSPLGEAC[Carb	BSK1-S353	phospho-peptide 1
	AT4G35230.1 [347-367]	QEEAPST[Phospho (STY)]PQRPLSPLGEAC[Carb	BSK1-S353	phospho-peptide 2
	AT4G35230.1 [228-242]	SYS[Phospho (STY)]TNLAYTPPEYLR	BSK1-S230	another phospho-site, not used here
	AT4G35230.1 [228-242]	S[Phospho (STY)]YSTNLAYTPPEYLR	BSK1-S228	another phospho-site, not used here
			BSK1-S353-sum	phospho-peptide-sum (1+2)
	positions in master proteins	Peptide used for quantification	ProteinName	
protein	AT4G35230.1 [322-333]	DLVATLAPLQTK	BSK1	protein 1
	AT4G35230.1 [228-242]	SYSTNLAYTPPEYLR	BSK1	protein 2
	AT4G35230.1 [68-85]	AATNFFSSDNIVSESGEK	BSK1	protein 3
			BSK1-sum	protein-sum (1+2+3)
Normalization				
			BSK1-S353/protein-sum	phospho-peptide 1/protein-sum
			BSK1-S353/protein-sum	phospho-peptide 2/protein-sum
			BSK1-S353-sum/protein-sum	phospho-peptide-sum/protein-sum

We further normalized the two BSK1-S353 phospho-peptides and their **sum (1+2)** to the **protein sum** of BSK1, respectively, which reached similar results. Now, we show the **phospho-peptide-sum/protein-sum** instead of the previous ‘phospho-peptide 1/protein-1’ in **Supplementary Fig.13** in the revised manuscript.

All the data and the detail of normalization can be found in the revised Supplementary Table 6.

Question 4:

For the normalization of the protein ratio, we work with a minimum of 3 peptides, where we always mixing the sequences of the non-phospho-peptide as well.

For the normalization of the total proteome, we use 3 background proteins, taking care that these proteins cannot be excessively modified. Therefore, histones are not so well suited.

Response: As *Reviewer 1* suggested, all the sequences of the protein (non-phospho-)peptide are now mixed in all the above analysis.

As *Reviewer 1* said, histones are not so well suited in the normalization of the total proteome. However, in this study, we are concerned with the phosphorylated H2A.X (γ -H2A.X) in *xip* mutant compared to WT, therefore, we normalized the phospho-

peptides by the corresponding proteins in this study.

Summary:

I would like to thank the authors for supplementary figure 12, which shows the comparison of the selected phosphorylation modification identified by LFQ and PRM. Supplementary figure 12 support the conclusion from histone Western that γ -H2A.X level has been significantly increased in xip-BLM when compared to WT-BLM.

I would like to thank you very much for the re-analysis of the data. If the ptm-rs module was used in proteome discoverer 2.4 workflow, it should be cited accordingly.

Response: We thank *Reviewer 1* for the encouragement and suggestion. We have cited the reference of ptm-rs node (Taus et al., 2011/PMID: 22073976) in the revised manuscript.

Supplementary Reference:

Taus T, *et al.* Universal and confident phosphorylation site localization using phosphoRS. *J Proteome Res* **10**, 5354-5362 (2011).

Reviewer #2 (Remarks to the Author):

This is an extensively revised version of a previous paper.

The authors performed a search for Arabidopsis proteins that specifically interact with a peptide based on the phosphorylated version of H2AX (and not the non-phosphorylated peptide). They come up with a single gene that has not been previously described and proceed to investigate some activities of the protein (its localization, its interactions with other proteins) and many aspects of the phenotype of a KO mutant, including effects on the transcriptome and phosphatome. There's a lot of data here, which is great, but there still are many issues with interpretation, and lesser issues with description of the methods and presentation of the data. The phenotype is subtle and it is difficult to draw any conclusions as to what this protein actually does. Conclusions are often underthought and overstated (and therefore misleading to many readers). The authors, in the previous version, felt it was a MDC1 homolog, and they may have changed their minds in this version, but there's still a lot of vestigial discussion of MDC1.

In mammals, MDC1 interacts with phosphorylated gH2AX and seems to be required for the "interpretation" of this signal- the phenotype of the MDC1 KO is very similar to that of an H2AX KO. But this protein, XIP, does not resemble MDC1 (except in its dual BRCT domains, and many proteins with various repair- or response related functions, have this dual BRCT domain). In their "rebuttal" the authors state that they regard this protein as a "functional analog" of MDC1- though they don't actually state this in the paper (in fact, they continue to imply it is a homolog), nor do they provide any evidence suggesting that this protein performs the same function as MDC1 - other than its binding to gH2AX. In the KO mutant the damage signal still clearly gets through to the plant transcriptome. The authors claim that the response is slightly muted, but some of this "muting" effect on the number of genes induced or repressed by damage may be due to the constitutive, low level, induction of the DDR response in

even untreated mutant plants. The authors need to keep in mind that there are many, many ways to disrupt DNA metabolism- constitutive low level DNA damage response and genome instability can result from errors in nucleotide pool regulation, redox regulation, DNA replication, transcription, repair, and, finally, damage response. One mutator mutant (mutQ) in E coli is actually a mis-loading tRNA, so add errors of translation to that list too.

What I think we have here is a new, previously undescribed gene, encoding a protein that probably binds H2AX, probably only in its phosphorylated form, and also binds RAD51. Plants carrying a KO of XIP grow a little more slowly than wt, are slightly more sensitive to the effects of DNA damaging agents on growth, and have a constitutive but low level transcriptional damage response- suggesting there is a constitutively higher than normal level of endogenous DNA damage. There's no reason to conclude that this is a response defect rather than a repair defect.

I have no expertise in phosphopeptide analysis, so I'm not discussing that.

My summary is that this is no doubt a gene that is somehow involved in DNA repair or response, but although the authors have put in an exceptional effort, we really have no idea of what this gene does. As the authors state in their rebuttal, there's really no place to put this gene in a diagram of the response or repair pathways.

Here's more detailed analysis of what I still find unconvincing (in terms of the data), and what I find "solid". Plus suggestions on English, which I think is mostly understandable, though it could use some cleaning up.

Response: We thank *Reviewer 2* for the suggestions.

L35 clarify whether you mean the response is depressed or enhanced, if you have a conclusion.

Response: We thank for the suggestion. The text has been revised in the Abstract.

L39 The conclusion that XIP is required to relay the H2AX signal is in my opinion completely unjustified. Again, as I stated in my last review, a mutant required to relay the signal would match the phenotype of a mutant completely defective in gH2AX- the authors don't do this experiment. See also line 100.

Response: We thank for the suggestion. The text has been revised in the Abstract.

L139 The authors again use the term "insignificant" and tell me in the rebuttal that that's what the Pfam program labeled it- they still don't define it. Please let us know what Pfam is trying to tell us with this term. It's a weird way to describe a domain that's essential for the one function that you know this protein performs- binding of your peptide.

Response: We have clarified the 'significance' in the revised manuscript. It represents the similarity of target domain sequence with the founding domain (RTT107_BRCT_5/PF16770 or BRCT/PF00533) identified by Pfam. Notably, the 'significance' or 'insignificance' is defined and labelled by Pfam but not the authors.

L143 "is close to...MDC1" is a meaningless term- you've already told us your domain is a dual BRCT, and as mentioned by another reviewer, it is "closer" the dual domains of other human proteins. I would stop trying to cram this protein into the MDC1 homolog/analog box and view it with fresh eyes for what it is- a protein of unknown function.

Response: We have revised the text.

L149 “unexpectedly low similarity in primary protein sequence may explain why clear MDC1 homologs have not been found in previous plant studies”. This outrageous statement clearly states that you have found the homolog in THIS study, which you have not. There is zero homology outside the BRCT domains, which are a frequently-employed motif.

Response: We are expressing the fact that the homolog of animal MDC1 has never been found in plants. The lack of sequence homology of full-length proteins may be the reason. In addition, the certain dual-BRCT-like domain (RTT107_BRCT_5/PF16770 followed by BRCT/PF00533) in XIP, which has been proved to contribute to the specific interaction with γ -H2A.X, has not been found in any plant proteins with reported functions in the previous studies.

We regret if our statement has given rise to ambiguity. We thank for the suggestion and have deleted the sentence in the revised manuscript.

L178 What on earth is going on with the cytoplasm/vacuole in the treated protos? Has the vacuolar membrane been lost? Is it always like this after treatment and never like this in untreated protos? If the cytoplasm and vacuole are mixed, the signal might simply be two dilute to see. So it's is not clear that the protein has migrated or simply been diluted in this vacuolar/cytoplasmic mix.

Response: Nothing has happened to the cytoplasm/vacuole and the vacuolar membrane has not been lost. Protoplasts without the constrain of intact cell wall resemble perfect spheroids, with nuclei and organelles pushed to one side due to the presence of pressure by vacuoles. Protoplast is a 3-D sphere. However, confocal observation can only provide 2-D images, the different observation angle chosen will lead to results with slight difference. When the nucleus and organelles are exactly

located in the focus plane during confocal observation, we can see the image as ‘Mock’ in Supplementary Fig.5. If they are not in the same focus plane (since the protoplast is a sphere, it can freely rotate), we can see the image as ‘BLM’ in Supplementary Fig.5 (in this image, nuclei and organelles are located above the focal plane), and as a result, the morphology of whole vacuole cannot be directly seen. Regretfully, *Reviewer 2* thought that vacuoles were mixed with cytoplasm. If this is the case, all organelles and nuclei should be freely scattered throughout the protoplast, rather than being visibly pushed to one side of protoplast. To support this, we provide another picture of ‘BLM’ by different view (Supplementary Information 1), in which the nucleus and organelles located exactly in the focal plane, once again confirming that the vacuole membrane has not ruptured.

Supplementary Information 1

Note: The dotted white ring represents the virtual plane where the nucleus (yellow) and protoplast center are located. In the original figure (middle BLM), the white ring is not overlapped with the focal plane, thus causing the nucleus and organelles to deviate from the focus plane.

We keep the original Supplementary Fig.5 in the revised manuscript.

L185 Fig. 1e does show one photo in which the two labels overlay- though the foci are not very obvious. Please also show us an untreated cell.

Response: We provide the immune-staining observation of an untreated protoplast as **Supplementary Fig.6** in the revised manuscript.

Note: The immuno-staining of an untreated protoplast was performed as a negative control in parallel to test the validity of our γ -H2A.X antibody. Here, we cannot find any γ -H2A.X signal. Different from the fluorescent observation in intact cells in **Supplementary Fig.5**, the immuno-staining protocol involves necessary membrane breakage for antibody to enter the cell as well as extensive washes. Probably for this reason, only nuclear YFP-XIP signal were clearly observed.

L187 I agree with everything in this paragraph! But please clarify the nature of the treatment- for example, how long were the plants exposed to the damaging agent? Provide enough detail for another lab to repeat your experiment.

Response: If *Reviewer 2* refers to the L187-corresponding paragraph (and thus the whole Figure 2), our answer is that these treatments were long-term. The plants were germinated and grown on MS medium with or without HU/BLM, which is very common in the phenotype observation in plant DDR pathway analysis. To make it clearer, we have changed “grown on” to “germinated and grown on” in the figure legend of **Figure 2**.

L212 again, we don't know the duration of the treatment

Response: We have clearly written the duration of HU and BLM treatment used for transcript and transcriptome analysis in the **Methods**. *Reviewer 2* can find the details of genotoxin treatment in the section of “**Transcript and Transcriptome analysis**” in **Methods**. To satisfy *Reviewer 2*, we have added the detail of genotoxin treatment in every figure legend in the revised manuscript.

L215 drop "much"- that's meaningless, especially as this was significant only at <0.05

Response: We thank for the suggestion.

L221 replace “all the examined” repair genes with “all three examined”

Response: In our transcript analysis, there are four BLM-activated DNA repair genes, *WEE1*, *PARP2*, *RAD51* and *RAD54*, and all these examined genes were not fully activated in *xip* mutant as compared to WT (**Fig.3b**, there are 4 “X” in the chart). In the figure legend, “X” indicates statistically significant difference in *xip*-BLM when compared with WT-BLM sample.

L236 You’re comparing the number of genes induced at least 2x, with a certain P value, in treated vs untreated mutants. Your xip control/wt control data already tells you that you have low-level constitutive expression of DDR in xip untreated (which is the most convincing proof that these mutants are experiencing constitutive genomic stress). But this might also explain why your treated xip plants do not have as many new genes induced by damage as wt plants, because they are already overexpressing these genes to some degree, so their fold induction is less significant.

Response: We thank for the suggestion, and changed the text in the revised manuscript.

L282 Quantitation of gH2AX is useful but you are leaping to a single conclusion- not considering alternative hypotheses. More gH2AX may be present because:

There are more breaks, as you suggest, but also, instead...

gH2AX is more persistent at each break (it is not being unloaded after repair)

gH2AX is more extensively loaded at each break

and you can't distinguish between more breaks generated by your DNA damaging agent (which you suggest but frankly seems unlikely? what would be the mechanism?)

breaks are persisting longer, i.e. slow repair

Response: First, the hypothesis raised by *Reviewer 2* is about the regulation of DSB-associated γ -H2A.X kinetics, including the phosphorylation and dephosphorylation of H2A.X as well as their dynamic equilibrium.

Second, we do not quite understand the question “*gH2AX is more extensively loaded at each break*”. Actually, γ -H2A.X is the phosphorylated form of H2A.X. If the question points to the exchange of H2A with H2A.X, or the DSB-induced loading of histone variant H2A.X around the DSB site, it is quite a big story far beyond this study.

Unless a plant research platform has been well established to detect the kinetic parameters of γ -H2A.X near a single DSB (especially, such DSB site should be extremely specific and constant) at the cellular level, we cannot test the possibility of the hypothesis proposed by *Reviewer 2*. Our Western blot experiment was performed by using WT and *xip* seedlings (but not cells) grown with or without BLM treatment. Obviously, such experiment cannot answer *Reviewer 2*'s *hypotheses*.

To satisfy *Reviewer 2*, we changed the corresponding statement in the revised manuscript.

L312 I'm not a fan of the comet assay for anything but the most obvious differences- especially in plants, and especially when they look like comet "hedgehogs". I think the transcriptional response is more convincing as evidence of instability. They might also consider a sectoring assay for chromosome arm loss.

Response: Comet assay is widely used to detect DNA damage in plant DDR research, for instance, the functional analysis of *SOG1/WEE1/RAD51* genes mentioned in this study (Cools et al., 2011/PMID: 21498679, Wang et al., 2014/PMID: 24102485, Yoshiyama et al., 2017/PMID: 29208704). However, *a sectoring assay for chromosome arm loss* has been rarely used to examine plant genome stability. It may be powerful in plant meiosis analysis, but it is difficult for us to employ such a technique when studying the heterogeneous somatic cells in multicellular plant Arabidopsis.

Supplementary Reference:

Cools T, et al. The Arabidopsis thaliana Checkpoint Kinase WEE1 Protects against Premature Vascular Differentiation during Replication Stress. *Plant Cell* **23**, 1435-1448 (2011).

Wang Y, et al. The Arabidopsis RAD51 paralogs RAD51B, RAD51D and XRCC2 play partially redundant roles in somatic DNA repair and gene regulation. *New Phytol* **201**, 292-304 (2014).

Yoshiyama KO, Kaminoyama K, Sakamoto T, Kimura S. Increased Phosphorylation of Ser-Gln Sites on SUPPRESSOR OF GAMMA RESPONSE1 Strengthens the DNA Damage Response in Arabidopsis thaliana. *Plant Cell* **29**, 3255-3268 (2017).

L322 I do like the PI assay for cell death but the authors need to tell us how they

quantified this. They again show us single photos and draw remarkable conclusions from images that look very similar (1 dead cell in a hundred is not significantly different from zero dead cells in a hundred). The images do have a tiny notation in the corner, for example, 20/20 or 17/20. They need to tell us what these mean- I'm guessing they just scored for presence or absence of dead cells, and didn't count the frequency of dead cells in each root tip, but help me out here.

Response: We thank for the suggestion. The sporadic cell death appeared under mock growth condition can be observed only in *xip* mutant (18/20 in Fig.5a and 17/20 in Supplementary Fig.14b), but not in any other plants (all WT and *XIP* in *xip* root tips).

After BLM treatment, obvious cell death can be found in all the root tips, except those of *sog1-101* and *sog1-101 xip*. We counted the number of dead cells in each root tips pictured by PI-staining and showed this result in **Supplementary Fig.15** in the revised manuscript. The detail can also be found in the **source data**.

L376 again, misleading overstatement. XIP is not required for HR activity- it still occurs. It is somewhat reduced- that's different.

Response: We thank for the suggestion and changed the text in the revised manuscript.

L377 "The finding of decreased HRF was also consistent with the increased chromatin instability". Again, think of additional hypotheses- decreased somatic mutagenesis (here, the homologous repair of spontaneous breaks at the test locus) is ALSO consistent with DECREASED rates of spontaneous breaks. Both are true, so it's not a very helpful assay.

Response: We are sorry that we do not understand the *additional hypotheses- decreased*

somatic mutagenesis / DECREASED rates of spontaneous breaks and why the HRF assessment is *not a very helpful assay* in plant study. To our knowledge, ‘somatic mutagenesis’ is largely correlated to human/animal aging and cancer. Based on our finding that XIP can interact with key recombinase RAD51, we performed HRF assessment of 12-day-old WT and *xip* in parallel in different concentration of exogenous DSB-inducing BLM (0, 2.5 and 10 μ M). The decreased frequency of HR, the critical DSB repair pathway, is consistent with the higher instability in *xip* chromatin in face of genotoxin stress, which has been further convinced by our other results, including transcription analysis, histone Western blot, comet assay and PI staining.

L383 why “unique”? You hadn’t mentioned “uniqueness” before. Do you just mean “a gene no one has discussed before”? If so drop the term.

Response: We are sorry that we have mentioned the “uniqueness” in the original manuscript. Now, we have changed the text in the revised manuscript.

L386 again, "required for cell cycle progression" is a HUGE overstatement, both in terms of phenotype and specificity of function. The mutants are slightly smaller than wt. Cell cycle progression is still happening. There's no evidence that this gene is at all involved in cell cycle progression as opposed to virtually any housekeeping function required for optimal growth. Rethink this sentence.

Response: We have changed the text in the revised manuscript.

L388 again, you have no idea at all what the function of this gene is. It has an impact on DDR- that does not really mean its function is to regulate DDR. Virtually anything

that impacts DNA metabolism- i.e., affecting nucleotide pools, replication, transcription, repair, or response, will impact genome stability.

Response: We have changed the statement and added a model as Supplementary Fig.17 in the revised manuscript.

L444 please don't tell us about this mystery protein until you have published data on it.

Response: We have identified several XIP-interacting proteins through pulldown-MS analysis, and importantly, such interacting protein (At2g41450, with the highest score) has the same dual-BRCT-like domain as XIP, which raises the possibility of further protein-protein interaction with phosphoprotein(s). In the **Discussion**, this sentence is just a perspective of the future molecular and functional analysis, and this is a NOT a **mystery** protein. We hope to publish these results in the future.

To satisfy *Reviewer 2*, we have changed the text in the revised manuscript.

Reviewer #3 (Remarks to the Author):

I would like to thank the authors for successfully addressing most of my concerns. However, there are still several issues remaining.

Response: We thank *Reviewer 3* for the appreciation.

The plant nomenclature of gene and protein names uses capital letters for the WT variants and small letters for mutant versions (+ italics for genes and non-italics for proteins). Therefore, „ATM and Rad3-related“ should be „ATM AND RAD3-RELATED“. I recommend fixing this throughout the manuscript.

Response: We thank *Reviewer 3* for the suggestion. The gene and protein names have been checked throughout the revised manuscript.

I acknowledge the authors' response about their view on the analogy between XIP and MDC1. However, I still do not understand their absolute preference towards human MDC1 (mentioned 74 times in the manuscript) while entirely neglecting (0 mentions) PTIP and PAXIP with higher homology to BRCT domains of Arabidopsis XIP protein. Although the authors write that they weakened their statement, the current version of the manuscript still offers MDC1 as a single option (see discussion part „ γ -H2A.X recognition by XIP and MDC1“). In short, I do not deny that XIP is a functional analog of human MDC1, but I am still missing strong argumentation why this is the best option.

Response: In this study, we have blasted XIP-L with *Homo sapiens* (taxid:9606) XM/XP database, and we found only two targets, HsMDC1 and HsPAXIP (PAX-INTERACTING PROTEIN 1)/PTIP (PAX TRANSCRIPTION ACTIVATION

DOMAIN-INTERACTING PROTEIN) (BLAST score: 107 and 84.3, shown as below).
The homology is limited to the C-terminal dual-BRCT-like domain of XIP.

Description	Scientific Name	Max Score	Total Score	Query Cover	E value	Per. Ident	Acc. Len	Accession
[x] PAX-interacting protein 1 [Homo sapiens]	Homo sapiens	107	107	25%	8e-24	33.17%	1069	NP_031375.3
[x] mediator of DNA damage checkpoint protein 1 [Homo sapiens]	Homo sapiens	84.3	84.3	24%	1e-16	25.26%	2089	NP_055456.2

Unexpectedly, HsPAXIP/PTIP contains 5 BRCT-like domains, in which 2 dual-BRCT-like domains are formed. In some studies, an about 80-aa-long sequence after the C-terminus of 1st BRCT-like domain is also considered as a BRCT-like domain, thus forming the third dual-BRCT-like domain (e.g. Zhang et al., 2022/PMID: 36153541). In great contrast, HsMDC1 comprises only one dual-BRCT-like domain as Arabidopsis XIP (see below).

The N-terminal and the middle BRCT-like domains of PAXIP also interact with other phospho-epitope in many proteins, including the most-studied 53BP1 and PAX2 which

do not exist in plant, linking PAXIP activity to multiple aspects of the human cellular response to DNA damage (Lechner et al., 2000/PMID: 10908331, Munoz et al., 2007/PMID: 17690115). In contrast, the C-terminal dual-BRCT-like domain of HsMDC1 can specifically recognize C-terminal tail of γ -H2A.X. By using the highly conserved tail of plant γ -H2A.X, we identified XIP as a highly specific γ -H2A.X-interacting protein. As a result, in this study, we focused on the functional comparison of XIP with HsMDC1.

We have revised the manuscript, and added the introduction of HsPAXIP.

Supplementary Reference:

Lechner MS, Levitan I, Dressler GR. PTIP, a novel BRCT domain-containing protein interacts with Pax2 and is associated with active chromatin. *Nucleic Acids Res* **28**, 2741-2751 (2000).

Munoz IM, Jowsey PA, Toth R, Rouse J. Phospho-epitope binding by the BRCT domains of hPTIP controls multiple aspects of the cellular response to DNA damage. *Nucleic Acids Res* **35**, 5312-5322 (2007).

Zhang F, Wei M, Chen H, Ji L, Nie Y, Kang J. The genomic stability regulator PTIP is required for proper chromosome segregation in mitosis. *Cell Div* **17**, 5 (2022).

As mentioned in the first round of revisions, I am also missing a model or discussion about the position of XIP in the DSB repair pathway.

Response: Based on our findings in this study, we set up a model about XIP in DSB repair pathway as **Supplementary Fig.17** in the revised manuscript.

Note: the added numbers are only shown in the *response* to highlight the corresponding experimental support in this study, and are not shown in the **Supplementary Fig.17**.

- ① ppd-MS (Fig.1a and 1b), ppd-pulldown (Supplementary Fig.3), immunostaining (Fig.1e)
- ② pulldown-MS, pulldown (Fig.5c), co-IP (Fig.5d), BiFC (Supplementary Fig.16)
- ③ HRF assay (Fig.5e and 5f)
- ④ Comet assay (Supplementary Fig.14a)
- ⑤ transcriptional analysis (repair genes, Fig.3b)
- ⑥ histone γ -H2A.X Western (Fig.4a)
- ⑦ Comet assay (Supplementary Fig.14a)
- ⑧ phenotype (Fig.2, Supplementary Fig.8), transcriptional analysis (phase-specific genes, Fig.3a and 3b)

⑨ transcriptome analysis (Fig.3c and 3d, Supplementary Fig.9, 10, 11, 12)

⑩ PI-staining (Fig.5a, Supplementary Fig.14b and 15)